# Incorporation of the equilibrium temperature approach in a Soil and Water Assessment Tool hydroclimatological stream temperature model

**Xinzhong Du[1]; Narayan Kumar Shrestha[1]; Darren L. Ficklin[2]; Junye Wang[1]***

**1 Athabasca River Basin Research Institute (ARBRI), Athabasca University, 1 University Drive, Athabasca, Alberta T9S 3A3, Canada**

**2 Department of Geography, Indiana University, Bloomington, Indiana, USA**

**\*Correspondence to:**   junyew@athabascau.ca

**Abstract**

Stream temperature is an important indicator for biodiversity and sustainability in aquatic ecosystems. The stream temperature model currently in the Soil and Water Assessment Tool ( SWAT) only considers the impact of air temperature on stream temperature, while the hydroclimatological stream temperature model developed within SWAT model considers hydrology and the impact of air temperature in simulating the water-air heat transfer process. In this study, we modified the hydroclimatological model

by including the equilibrium temperature approach to model heat transfer processes at the water-air interface, which reflects the influences of air temperature, solar radiation, wind speed and streamflow condition on the heat transfer process. The thermal capacity of the streamflow is modelled by the variation of the stream water depth. An advantage of this equilibrium temperature model is the simple parameterization, with only two added parameters to model the heat transfer processes. The equilibrium

temperature model proposed in this study is applied and tested in the Athabasca River Basin (ARB) in Alberta, Canada. The model is calibrated and validated at five stations throughout different parts of the ARB where close to monthly samplings of stream temperatures are available. The results indicate that the equilibrium temperature model proposed in this study provided better and more consistent performances for the different regions of the ARB with the values of the Nash-Sutcliffe Efficiency coefficient (NSE)

greater than those of the original SWAT model and the hydroclimatological model. To test the model performance for different hydrological and environmental conditions, the equilibrium temperature model was also applied on the North Fork Tolt River Watershed in Washington, United States. The results indicate a reasonable simulation of stream temperature using model proposed in this study with minimum relative error values compared to other two models. However, the NSE values were less than those of the

hydroclimatological model indicating more model verification needs to be done. The equilibrium temperature model uses existing SWAT meteorological data as input, can be calibrated using fewer parameters and less effort, and has an overall better performance for stream temperatures simulation. Thus, it can be used as an effective tool for predicting the change in stream temperature regimes under varying hydrological and meteorological conditions. In addition, the impact of the stream temperature

simulations on chemical reaction rates and concentrations was tested. The results indicate that the improved performance of the stream temperature simulation could significantly affect chemical reaction rates and the simulated concentrations and the equilibrium temperature model could be a potential tool to model stream temperature for water quality simulations.

**Keywords:** SWAT model, stream water temperature, equilibrium temperature, Athabasca River Basin

**1. Introduction**

Stream temperature is an important factor in assessing water quality and biodiversity health. Stream temperature can alter physical and chemical properties of water bodies. It has effects on water density, conductivity, pH, dissolved oxygen concentration, compound toxicity, chemical reaction rates, biological activity and biological habitats. All aquatic species have a specific range of water temperature that they can tolerate and any changes in water temperature may have an adverse impact on the habitat of aquatic species (Caissie et al., 2007). For example, a fish species in a stream is likely to migrate if the maximum weekly stream temperature exceeds its temperature tolerance (Eaton et al., 1995). Also, a fish species could perish due to osmoregulatory dysfunction if weekly stream temperature drops below a threshold temperature (Mohseni et al., 1998). Stream temperature regimes have been and will continue to be affected by anthropogenic activities, including thermal inputs from industry and power plants and landuse and climate change. Land use changes, such as deforestation and urbanization, have an impact on watershed hydrological conditions that can lead to stream temperature changes (Cao et al., 2016). Moreover, the possible rise of water temperature due to global warming caused by climate change is expected to affect aquatic species directly and indirectly (Hardenbicker et al., 2017;Ducharne, 2008; Knouft and Ficklin, 2017). Therefore, it is important to model stream temperature for predicting the changes in temperature under varying hydrological and meteorological conditions.

Many stream temperature models have been developed over the past years, which can be classified into mechanistic and statistical models. A mechanistic model is based on energy balance while a statistical model uses regression techniques between stream temperature and meteorological or other physical variables. Widely used statistical models of stream temperature are regressed linearly or nonlinearly using only air temperature as an input parameter. Stefan et al. (1993) used a linear model between the stream and air temperatures with time lags to simulate daily and weekly water temperatures in 11 streams in the Mississippi River basin in the central US. Mohseni et al. (1998) developed a four-parameter nonlinear function using air temperature as the input to model weekly stream temperatures based on temperatures recorded over a 3-year period (1978-1980) at 584 U.S. Geological Survey gauging stations in the contiguous United States. Sohrabi et al. (2017) developed a parsimonious Bayesian regression approach to model daily stream temperatures accounting for the temporal autocorrelation, linear and nonlinear relationships with air temperature and discharge. However, stream temperature is clearly subject to other meteorological and hydrological constraints, such as solar radiation, wind speed and water depth (Neitsch et al., 2011), which cannot be reflected by the simple regression approach. Moreover, the impact of watershed hydrological conditions are not included in these regression models (Ficklin et al., 2012). Therefore, the statistical models of stream temperature may not be reliable when interpreting and predicting the impact of environmental and anthropological drivers, such as climate and landuse change.

Mechanistic stream temperature models simulate the change of stream temperature based on energy balances of heat fluxes and water mass balance in a river system (Brennan, 2015). Heat transfer and stream temperature are calculated at the water-air interface and water-sediment interface. Heat exchange between sediment and water is generally small compared to water-air heat exchange (Caissie et al., 2007) and can often be negligible. At the water-air interface, heat flux can be calculated using solar radiation, net long-wave radiation, evaporation, and convective heat transfer. As stream temperatures impact the chemical reaction rates in the aquatic environment, widely used water quality models, such as QUAL2K (Chapra et al., 2012) and CE-QUAL-W2 (Cole et al., 2016), have the capability to model stream temperature based on a full energy balance approach. These models require hydrological conditions represented by flow rates and stream temperature from tributaries as input boundary conditions to model stream temperature in the mainstream based on a full energy balance approach. Therefore, these algorithms need to be directly linked or implemented in hydrological models to assess the effects of watershed hydrological conditions on stream temperature. Previous work has tried to incorporate physical

based energy balances into hydrological models to simulate stream temperature. For example, Ozaki et al. (2008) developed a river temperature model based on a multi-layer mesh-type runoff model to calculate the heat budget in the landscape and river system. Battin et al. (2007) added a heat balance module to the Distributed Hydrology-Soil-Vegetation Model (DHSVM) to model stream temperatures in the Snohomish Basin. However, because of their complexity, these mechanistic models require intensive data and calibration effort, which to some extent, can limit their applicability (Du et al., 2014). Complex mechanistic models, however, might be more reliable by more realistically representing the physical processes compared to the statistical models. Therefore, the equilibrium temperature approach, which can be seen as a compromise between an empirical statistical and a complex mechanistic model, can be used as an alternative for simulating the heat transfer processes.

Ficklin et al. (2012) developed a hydroclimatological stream temperature model within the Soil and Water Assessment Tool (SWAT) hydrological model (Arnold et al., 1998), which includes the combined effects of watershed hydrological conditions and air temperature. It is a mechanistic model with a simplified representation of temperature mixing from different runoff components and water-air heat transfer processes. This model was tested and validated in seven coastal and mountainous basins in the western U.S., showing much better stream temperature simulation performance compared to the original linear regression approach of SWAT (Ficklin et al., 2012) . It has also been used to assess the impact of climate change on stream temperature in the Columbia River basin in North America (Ficklin et al., 2014) and Sierra Nevada mountain range in California (Ficklin et al., 2013). Zeiger et al. (2016) applied the hydroclimatological model in a mixed-use, urbanizing watershed in the central U.S. and compared the model performance with the linear and non-linear regression models. Their results showed that it had a better and more consistent performance both in lower and higher stream temperature ranges. The hydroclimatological model explicitly describes the effects of hydrological inputs (local runoff components and upstream inflow) on stream temperature for evaluating the impact of hydrological changes caused by climate or landuse changes on stream temperature. However, the process of water-air heat transfer is modelled by considering only the impact of air temperature. The water-air heat transfer can be simulated based on the full energy balance, but more input data and calibration will be required because of the model complexity. The equilibrium temperature approach, which includes the impact of air temperature, solar radiation, and wind speed and stream water depth (Bogan et al., 2003;Mohseni and Stefan, 1999) is an alternative for simulating heat trasfer processes. Although many studies have used the equilibrium temperature concept to interpret thermal processes in rivers, it has rarely been used for the simulation of stream temperature (Caissie et al., 2005). The equilibrium temperature approach is a compromise between an empirical statistical and a complex mechanistic model. It has moderate data input requirements and has potential to be an effective modelling tool for stream temperature. Therefore, in this work the equilibrium temperature approach is incorporated into this hydroclimatological model to improve the simulation of heat transfer process at the water-air interface.

The primary objective of this paper is to improve the simulation of the heat transfer process in the SWAT hydroclimatological stream temperature model by incorporating the equilibrium temperature approach. The equilibrium temperature model uses existing SWAT meteorological input data and a simple parameterization scheme with only two added parameters to model heat transfer processes at water-air interface. It not only includes the effects of air temperature, solar radiation and wind speed, but also incorporates variations in thermal capacity of streamflow represented by stream water depth. The SWAT model is calibrated for hydrology and the equilibrium model is then calibrated and validated against observed daily stream temperature data at five different stations throughout the upper, middle and lower regions of the  Athabasca River Basin (ARB), located in Alberta and Saskatchewan, Canada. To test the model performance for different hydrological and enviromental conditions,  the equilibrium

model was also tested on the North Fork Tolt River Watershed in Washington, United States, which is one of the seven watersheds used for the hydroclimatological model initial testing. The simulations of stream temperature impact the water quality concentrations by directly impacting the chemical reaction rates and therefore another objective of this paper is to initially evaluate the impacts of stream temperature simulations on water quality concentration modeling using original SWAT, hydroclimatological and equilibrium stream temperature models.

## 2. Materials and Methods

### 2.1 SWAT hydrological and stream temperature model

Meteorological data and spatial datasets are required for setting up the SWAT model in ARB. The spatial datasets include a digital elevation model (DEM), landuse data and soil data. The Shuttle Radar Topography Mission (SRTM) DEM data (90m×90m), Global Land Cover Characterization based land use map of 1km × 1km spatial resolution (Loveland et al., 2000) and 1:1 million scale soil map from the Agriculture and Agri-Food Canada were used as model input. The DEM was used to delineate subbasin and stream networks, where a total of 131 subbasins were delineated in the ARB. Eleven different landuse classes and 320 different soil types were defined for the model setup. A total of 1370 HRUs were defined based on the landuse, soil and slope classifications. To define the HRUs, a slope map is derived from the DEM and divided into 4 classes (breaks at 5%, 10%, 15% and 20%). Moreover, a threshold of 10%, 5% and 10% for land use, soil and slope, respectively, were used for defining HRUs. For meteorological data input, daily precipitation, maximum air temperature and minimum air temperature were obtained from 73 stations recorded by Environmental Canada and Climate Change. Relative humidity, solar radiation, and wind speed data at 230 stations recorded by Climate Forecast System Reanalysis (Dile and Srinivasan, 2014) were also used as the model input data.

The SWAT model is a river basin or watershed scale model used to predict the impact of land management practices on water, sediment and agricultural chemical yields in large complex watersheds with varying conditions over long periods of time (Neitsch et al., 2011). The simulated processes of the hydrological cycle in SWAT include canopy interception, surface runoff, infiltration, lateral flow, snowmelt flow, evapotranspiration, deep percolation, groundwater flow and water routing in the stream and other water bodies. Snowpack accumulation and snowmelt processes are modelled using a temperature index-based approach. More detailed description of SWAT theory is available from Neitsch et al. (2011).

The SWAT model uses a linear equation developed by Stefan and Preudhomme (1993) to calculate average daily temperature for a well-mixed stream:

$$T_w = 5.0 + 0.75 \cdot T_{air} \tag{1}$$

where $T_w$ is the stream temperature for the day (°C), and $T_{air}$ is the average air temperature on the day (°C). This equation assumes that the lag time between air and stream temperatures is less than one day. However, aside from air temperature, the stream temperature is influenced by other factors, such as solar radiation, wind speed, relative humidity, water depth, artificial heat inputs, groundwater inflow and thermal conductivity of the sediments. The impacts of these factors on stream temperature are not taken into account in the existing SWAT model versions. Therefore, this simple linear equation may lead to unrealistic estimates of stream water temperature when the air temperature is low during winter in cold regions like the ARB. Moreover, it might provide unrealistic simulations when air temperature suddenly drops or rises.

## 2.2 SWAT hydroclimatological stream temperature model

The hydroclimatological stream temperature model developed by Ficklin et al. (2012) is used to simulate the combined impacts of air temperature and hydrological inputs (streamflow, snowmelt, groundwater, surface runoff, and lateral soil flow) on stream temperature. Three components are considered in this stream temperature model, including temperature and water contribution within the subbasin, the temperature and volume of inflows from upstream subbasin(s) and heat transfer at the water-air interface during the streamflow transport in the subbasin. In the first step, the temperature of the local water contribution is calculated within the local subbasin using a basic mixing model of the volumes and temperatures of surface runoff, lateral flow, and groundwater, and snowmelt runoff to the stream water:

$$T_{w,local} = \frac{(T_{snow} \cdot sub\_snow) + (T_{gw} \cdot sub\_gw) + (\lambda T_{air,lag})(sub\_surq + sub\_latq)}{sub\_wyld} \tag{2}$$

where $sub\_snow$ is the snowmelt runoff contribution to streamflow within the subbasin ($m^3 d^{-1}$), $sub\_gw$ is the groundwater flow contribution to streamflow within the subbasin ($m^3 d^{-1}$), $sub\_surq$ is the surface runoff contribution to streamflow within the subbasin ($m^3 d^{-1}$), $sub\_latq$ is the soil lateral flow contribution to streamflow within the subbasin ($m^3 d^{-1}$), $sub\_wyld$ is the total water yield contribution to streamflow within the subbasin ($m^3 d^{-1}$), $T_{snow}$ is the temperature of snowmelt runoff (0.1°C ), $T_{gw}$ is the groundwater flow temperature (°C), $T_{air,lag}$ is the average daily air temperature with a lag (°C), and $\lambda$ is a calibration coefficient. The lag (days) is a parameter incorporated to allow the effects of delayed surface runoff and soil water flow into the stream. $\lambda$ is a calibration coefficient relating the relationship between $T_{air,lag}$ and surface runoff and lateral flow.

In the second step, the initial stream temperature before calculating heat transfer between air and water is then calculated as a weighted average of contributions within the subbasin and the contribution from the upstream subbasin(s):

$$T_{w,initail} = \frac{T_{w,up}(Q_{outlet} - sub\_wyld) + T_{w,local} \cdot sub\_wyld}{Q_{outlet}} \tag{3}$$

where $T_{w,up}$ is the temperature of the streamflow entering the subbasin from upstream subbasin(s) and $Q_{outlet}$ is the streamflow discharge at the outlet of the subbasin. In the case of headwater streams without inflow, $T_{w,up} = T_{w,initial}$.

In the third step, the final stream temperature is calculated by adding a change caused by heat transfer to the initial stream temperature. This change is calculated based on the temperature difference between the stream and air, and the water travel time through the reach in the subbasin. It is given by the following equations, depending on $T_{air}$:

$$T_w = T_{w,initial} + (T_{air} - T_{w,initial}) \cdot K \cdot TT \qquad\qquad \text{if } T_{air} > 0 \tag{4}$$

$$T_w = T_{w,initial} + [(T_{air} + \varepsilon) - T_{w,initial}] \cdot K \cdot TT \qquad\qquad \text{if } T_{air} < 0 \tag{5}$$

where $T_{air}$ is average daily temperature, K (1/h) is a bulk coefficient of the heat transfer and ranges from 0 to 1, TT is the water travel time in the stream (hour) and is simulated by the SWAT stream routing module, and $\varepsilon$ is air temperature addition coefficient, which is included to account for water temperature pulses when air temperature is below 0 °C. $\varepsilon$ allows the simulated stream temperature to rise above 0 °C when air temperature is below 0 °C. K is the critical parameter for calculating the heat transfer, which is

dependent on the relationship between stream and air temperature within a subbasin. For example, if stream temperature is approximately the same as air temperature, then K is 1. If there is a short travel time or extensive tree shading, then K will be less than 1 but greater than 0. K as 0 means there is no heat transfer between air and water.

## 2.3 Incorporating the equilibrium temperature approach into Ficklin et al. (2012)

The same equations (equations 1 to 3) were used to calculate the initial stream temperature in the reach. However, the equilibrium temperature approach was incorporated to simulate the heat transfer process. The change of stream temperature can be modelled based on an energy balance accounting for the heat exchange between water-air and water-sediment interface. Stream temperature increases or decreases with time according to the net heat flux:

$$\rho_w C_{pw} \frac{\partial T_w}{\partial t} = \frac{q_{net}}{H} \tag{6}$$

where $\rho_w$ is the density of water (kg/m$^3$), $C_{pw}$ is the specific heat capacity of water, $q_{net}$ is the net heat flux (W/m$^2$) and H is the water depth (m), which is calculated by the SWAT stream routing module.

The equilibrium temperature is defined as a hypothetical water temperature at which the net heat flux is zero. The net heat input is assumed to be proportional to the difference between the stream temperature and the equilibrium temperature:

$$q_{net} = K_T\left(T_e - T_w\right) \tag{7}$$

Combining equation 7 into equation 6 yields:

$$\rho_w C_{pw} \frac{\partial T_w}{\partial t} = \frac{K_T\left(T_e - T_w\right)}{H} \tag{8}$$

where $K_T$ is overall heat exchange coefficient (W/m$^2$/°C) and $T_e$ is the equilibrium temperature (°C).

The overall heat exchange coefficient can be calculated from the empirical relationships that include wind velocity, dew point temperature and initial stream temperature $T_{w,initial}$ (Edinger et al., 1974).

$$K_T = 4.5 + 0.05 T_w + \beta \cdot f(wnd) + 0.47 f(wnd) \tag{9.a}$$

$$f(wnd) = 9.2 + 0.46 wnd^2 \tag{9.b}$$

$$\beta = 0.35 + 0.015\left(\frac{T_d + T_w}{2}\right) + 0.0012\left(\frac{T_d + T_w}{2}\right)^2 \tag{9.c}$$

where $T_d$ is the dew point temperature (°C), wnd is the wind speed (m/s), which is an input meteorological data of the SWAT model. The equilibrium temperature can be calculated by the empirical relationship of the overall heat exchange coefficient, the dew point temperature and the solar radiation (below) (Edinger et al., 1974):

$$T_e = T_d + \frac{slr}{K_T} \tag{10}$$

where slr is the solar radiation, which is also an input meteorological data of the SWAT model.

In equations 9 and 10, the dew point temperature is required when calculating the heat exchange coefficient and equilibrium temperature. Because the dew point temperature is not a meteorological input of the SWAT model, it can be estimated by air temperature and relative humidity using a simple linear equation or non-linear equation (Lawrence, 2005). Dingman (1972) also used air temperature and solar radiation to calculate the equilibrium temperature instead of using dew point temperature as equation 10. In this study, air temperature and an additive parameter ($T_{air}$+η) are used to replace the dew point temperature $T_d$ in equation 8 and 9, and therefore the dew point temperature is not required as an input data. Using air temperature and an additive parameter makes the equilibrium temperature linearly related to air temperature in our model, which is consistent with other work (Caissie et al., 2005; Bustillo et al., 2013) that use the equilibrium temperature approach for water temperature modeling. The η is an additive parameter representing the linear relationship between air temperature and the equilibrium temperature, which is subject to model calibration using observed stream temperature data. Therefore, the equilibrium temperature approach proposed here can calculate the water-air heat transfer using SWAT existing input data, which considers the impact of air temperature, solar radiation, wind speed and water depth. The final stream temperature is corrected using the equilibrium temperature of the influence of heat transfer to the initial stream temperature. Combining equations 9 and 10 into equation 6 yields:

$$T_w = T_{w,\,initial} + \frac{K_T\left(T_e - T_{w,\,initial}\right)}{\rho_w C_{pw} H} \cdot TT \tag{11}$$

where $T_w$ is the stream temperature after the water-air heat transfer calculation using the equilibrium temperature, $T_{w,\,initial}$ is the initial stream temperature by mixing water from upstream and the local subbasin. The equilibrium temperature $T_e$ is calculated in the below formula using the air temperature rather than the dew point temperature:

$$T_e = (T_{air} + \eta) + \frac{slr}{K_T} \tag{12}$$

where η is the additive parameter for the water-air heat transfer process. Heat exchange coefficient $K_T$ is calculated using equation 8 with ($T_{air}$+η) replacing dew point temperature $T_d$.

## 2.4 Evaluating the impact of the stream temperature simulation on water quality modelling

Stream temperature simulation has an impact on water quality modelling in SWAT since the water temperature affects chemical reaction rates and oxygen saturation concentration. Therefore, the impacts on chemical reaction rates are analyzed at first. The water quality module of SWAT uses an exponential equation to correct chemical reaction rates based on the simulated daily stream temperature (Neitsch et al., 2011) and the impacts on reaction rates are based on the equation below:

$$k(T) = k_{20} \cdot \theta^{T_w - 20} \tag{13}$$

where K(T) is the reaction rate at a local temperature ($d^{-1}$), $k_{20}$ is the reaction rate at 20 °C ($d^{-1}$), θ is temperature correction coefficient, and $T_w$ is water temperature simulated by SWAT model (°C). In addition, the impacts of stream temperature simulation on water quality concentration modeling are investigated by outputting and comparing the simulated concentration based on three different models using the same parameter.

## 2.5 Study area

The Athabasca River originates in the Rocky Mountains of Alberta and travels northeast across Alberta (Figure 1). The ARB includes the urban centers of Jasper, Hinton, Whitecourt, Athabasca and Fort McMurray before draining into Lake Athabasca. The entire ARB is approximately 159000 $km^2$, which is about 24% of Alberta's landmass. Forest is the dominating land cover accounting for about 82% of the whole basin area, and agriculture land (9.5%) stands at a second. Major activities in the basin include forestry, agriculture, tourism, pulp mills, coal mining, traditional oil and gas extraction, and oil sands mining. Within the ARB, fish species can be broadly grouped into two primary types: those tolerant of cold waters and those which require relatively warmer water temperatures (Wallace and McCart, 1984). The main fish species include walleye, lake whitefish, northern pike, and burbot (Lebel et al., 2011).

**Figure 1** Location of Athabasca River Basin with streamflow and stream temperature stations.

The equilibrium stream temeprature model was also applied on the North Fork Tolt River Watershed to test the model perforamce for differernt hydrological and enviromental conditions. The North Fork Tolt River Watershed is located in Washington, Uinted States with the drainage area of 105 km$^2$. The annual average air temeperature and precipitation are 8.1°C and 2331 mm, respectively. More detailed information can be found in Ficklin et al. (2012).

**2.6 Model calibration and validation**

The SWAT model was used for the simulation of streamflow and stream temperatures for a 34 year period, from 1980 to 2013. The first two years (1980-1981) were used as a warm up period to minimize the impact of initial conditions. The model calibration period was from 1990 to 2005 (16 years) including both wet and dry periods. The years of 1982 to 1989 and 2006 to 2013 were used for the model validation. SWAT-CUP (SWAT Calibration and Uncertainty Programs) and its SUFI-2 (Sequential Uncertainty Fitting Version 2) algorithm were used for streamflow calibration and validation (Shrestha et al., 2017). For streamflow, daily time series data from 35 stations collected from Environmental Canada and Climate Change were used for the hydrological calibration and validation. The streamflow was calibrated from upstream to downstream according to the locations of 35 flow gauging stations. The Nash-Sutcliffe Efficiency coefficient (NSE) was selected as the objective function for the model calibration. In addition, the coefficient of determination ($R^2$) and relative error (RE) were also used to evaluate the model performance. The definitions of NSE, $R^2$ and RE can be found in Du et al. (2016).

The same calibration and validation period were used for the stream temperature. Stream temperature data from Environmental Canada and Climate Change was used for the stream temperature calibration. The sampling frequencies of stream temperature varied from monthly to seasonal, and five stations with close to monthly sampling frequency were chosen for the model calibration. These five stations were located from upstream to downstream reflecting stream temperature conditions in different parts of the ARB (Figure 1). Additional information for the five stream temperature observed stations can be found in Table 1. As there was no stream temperature data at Athabasca River near Windfall during the validation period, model validation was performed at other four stations. The equilibrium and hydroclimatological stream temperature models were manually calibrated. To test the model performance and validity with less calibration effort, one set of the stream temperature parameters for the ARB were used for the calibration processes of the equilibrium stream temperature, which means that the same parameters were used for different subbasins and seasons. However, seasonally varied parameters were used for the calibration of the hydroclimatological stream temperature model. For the hydroclimatological stream temperature model, it is recommended to use different parameters for different seasons which can account for the influence of seasonal variation on the stream temperature. In this study, three periods were defined to represent seasonal variations and different parameters were given for each period. The average NSE value of the five stations was chosen as the objective function for the stream temperature calibration, and their parameter values were adjusted to obtain the maximum average NSE value. The average values of $R^2$ and RE were also calculated to evaluate the performance. The RE can be positive in case of overestimation or negative in case of underestimation. Therefore, the absolute value of RE was calculated for each station, and then these absolute values were averaged.

**Table 1** Detailed information for the five stream temperature stations in the Athabasca River Basin

| Stations | Drainage area (km$^2$) | Watershed Location | Number of water temperature samples | |
| --- | --- | --- | --- | --- |
| | | | Calibration | Validation |

| | | | |
|---|---|---|---|
| Athabasca river at Old Fort | 154800 | downstream | 97 | 71 |
| Muskeg River near Fort Mackay | 1715 | downstream | 102 | 39 |
| Athabasca River near Horse River | 98270 | downstream | 103 | 99 |
| Athabasca River at Athabasca | 73580 | midstream | 187 | 117 |
| Athabasca River near Windfall | 19650 | upstream | 60 | / |

## 3. Results and Discussion

### 3.1 SWAT streamflow calibration and validation

To examine the performance of the equilibrium temperature model, a SWAT model calibrated for hydrology is imperative in order to perform an accurate simulation of the stream temperature. The SWAT hydrological model was calibrated and validated using daily streamflow data for 35 stations in the ARB to ensure these stations can represent upper, middle and lower parts of the ARB (Figure 1). The average values of NSE for the 35 stations were 0.57 and 0.49, respectively, during calibration and validation period. The average values of RE were 5.3% and 12.4%, respectively, during calibration and validation period. Detailed results from the ARB SWAT streamflow calibration and validation can be found in Shrestha et al. (2017). Overall, the accuracy of streamflow results at 35 stations across the basin suggests that the SWAT model could simulate the streamflow at headwaters, foothills, and prairie regions reasonably well, and the model's accuracy at downstream parts of the boreal plain region was satisfactory (Shrestha et al., 2017). This well calibrated SWAT hydrological model is further used for the simulation of stream temperature based on the hydroclimatological and equilibrium temperature models.

### 3.2 Stream temperature calibration and validation

For the equilibrium temperature model, a range of 0.5 to 1.5 for the multiplicative factor parameter $\lambda_{kt}$ was used for calibration. $\eta$ was the most sensitive parameter for the equilibrium temperature model and therefore it was the main parameter that was manipulated. $\eta$ was the first parameter to be calibrated, and it was optimized to 3.2 to obtain the maximum average NSE value of 0.79. Other parameters (Lag, $\lambda$ and $\lambda_{kt}$) had little effect on NSE and $R^2$ value, but they did, however, have an impact on the average RE value. After $\eta$ was calibrated, Lag, $\lambda$ and $\lambda_{kt}$ were calibrated as 2, 1.1 and 1.15, respectively, to minimize the average RE value. In addition, it is important to give default parameter values to make the model more applicable and the default values can be used for the users who are not able to calibrate for stream temperature. The default parameter values are given as follows: $\eta$ as 0, Lag as 3 days, $\lambda$ and $\lambda_{kt}$ as 1.0. However, it is recommended to use observed stream temperature data to calibrate the model parameters instead of using the default values. For the hydroclimatological model, three different seasons (Table 2) with different parameter sets were used for model calibration. It is found that the heat transfer coefficient K is the most sensitive parameter for hydroclimatological model and was the main parameter calibrated. The calibrated parameter values are given in Table 2.

**Table 2** Calibrated parameters of the hydroclimatological stream temperature model for the Athabasca River Basin

| Julian Day | | $\lambda$ | K (1/h) | $\varepsilon$(°C) | Lag (days) |
|---|---|---|---|---|---|
| From | To | | | | |

| 1 | 90 | 1.1 | 0.1 | 12 | 5 |
|---|---|---|---|---|---|
| 91 | 300 | 1.1 | 0.05 | 0 | 3 |
| 301 | 366 | 1.1 | 0.1 | 9 | 5 |

### 3.3 Performance of the equilibrium temperature model

To evaluate the performance of the equilibrium temperature model, we performed a comparison of the daily model simulation results with observed stream temperature in the ARB (Figure 2). The results of
5 equilibrium temperature model in this study are in good agreement with the observed data for all five stations in the ARB. Furthermore, the simulated results of the equilibrium temperature model were compared with that of the original SWAT temperature model and that of the hydroclimatological stream temperature model. Table 3 shows the results of statistical performance for the three different stream temperature models. For all five stations, the equilibrium temperature model improved the performance of
10 stream temperature simulations compared to that of other two models. The original SWAT stream temperature model had average NSE, $R^2$ and RE as 0.51, 0.66 and 10.1%, respectively, during the calibration period and 0.56, 0.67 and 8.4%, respectively, during the validation period. The hydroclimatological model had average NSE, $R^2$ and RE as 0.50, 0.61 and 16.5%, respectively, during the calibration period and 0.50, 0.62 and 19.7%, respectively, during the validation period. The equilibrium
temperature model had as average NSE, $R^2$ and RE as 0.79, 0.82 and 9.6%, respectively, during the calibration period and 0.76, 0.80 and 7.4%, respectively, during the validation period. Furthermore, the equilibrium temperature showed good performances for all regions of the ARB, with NSE values all greater than 0.67. In contrast, the performances of other two models are not consistent among different regions, especially for Athabasca River Station near Windfall with NSE values less than 0.1.

**Table 3** Calibration and validation statistics for the three stream temperature models for the five stations in the Athabasca River Basin

| Stations | Different models | Calibration | | | Validation | | |
|---|---|---|---|---|---|---|---|
| | | NSE | $R^2$ | RE (%) | NSE | $R^2$ | RE (%) |
| Athabasca River at old Fort | Original SWAT model | 0.48 | 0.62 | 15.9 | 0.68 | 0.71 | 11.8 |
| | Hydroclimatological model | 0.68 | 0.68 | 1.5 | 0.81 | 0.87 | -15.7 |
| | Equilibrium temperature model | 0.67 | 0.74 | 14.6 | 0.70 | 0.75 | -0.8 |
| Muskeg River near Fort Mackay | Original SWAT model | 0.60 | 0.69 | 11.1 | 0.54 | 0.67 | 16.8 |
| | Hydroclimatological model | 0.36 | 0.46 | 18.3 | -0.03 | 0.30 | 47.0 |
| | Equilibrium temperature model | 0.81 | 0.85 | 12.0 | 0.80 | 0.86 | 18.8 |
| Athabasca River near Horse river | Original SWAT model | 0.52 | 0.62 | -14.0 | 0.50 | 0.60 | -4.8 |
| | Hydroclimatological model | 0.6 | 0.61 | -4.1 | 0.41 | 0.46 | -0.3 |

| | | | | | | | |
|---|---|---|---|---|---|---|---|
| | Equilibrium temperature model | 0.86 | 0.86 | -0.9 | 0.74 | 0.76 | 7.8 |
| Athabasca River at Athabasca | Original SWAT model | 0.87 | 0.68 | 2.8 | 0.53 | 0.68 | -3.5 |
| | Hydroclimatological model | 0.81 | 0.82 | -11.9 | 0.8 | 0.83 | -15.8 |
| | Equilibrium temperature model | 0.86 | 0.86 | -3.3 | 0.80 | 0.81 | -2.3 |
| Athabasca River near Windfall | Original SWAT model | 0.09 | 0.68 | -6.6 | / | / | / |
| | Hydroclimatological model | 0.03 | 0.47 | 46.9 | / | / | / |
| | Equilibrium temperature model | 0.74 | 0.77 | -17.2 | / | / | / |
| **Average** | Original SWAT model | **0.51** | **0.66** | **10.1** | **0.56** | **0.67** | **8.4** |
| | Hydroclimatological model | **0.50** | **0.61** | **16.5** | **0.50** | **0.62** | **19.7** |
| | Equilibrium temperature model | **0.79** | **0.82** | **9.6** | **0.76** | **0.80** | **7.4** |

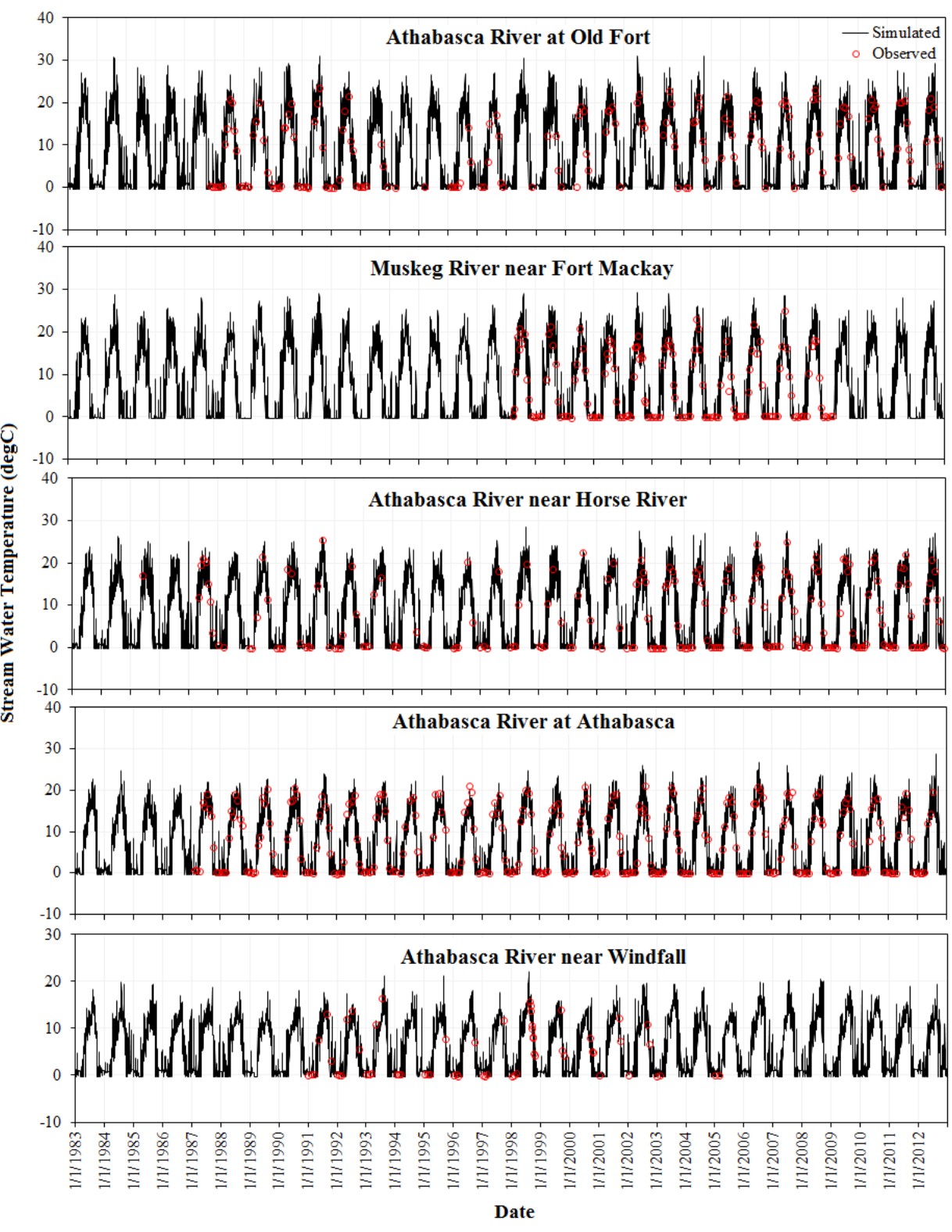

**Figure 2** Comparisons of simulated stream temperature using the equilibrium temperature model with observed stream temperature in the ARB.

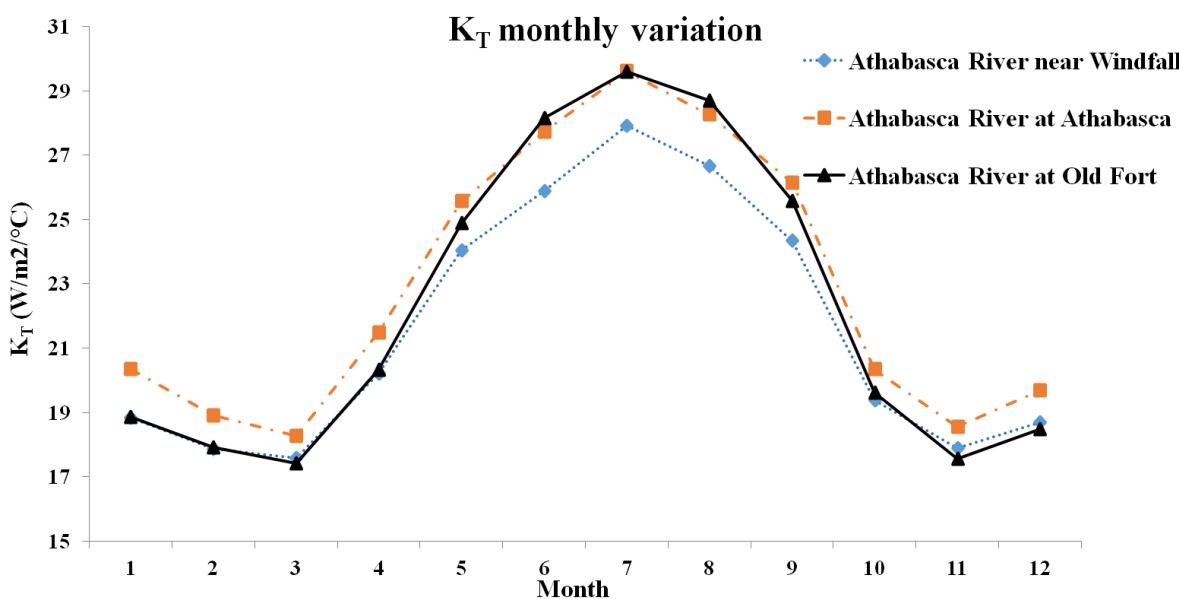

**Figure 3** Monthly variations for the heat exchange coefficient $K_T$ at different regions of ARB

Another advantage of the equilibrium temperature model is the simple parameterization scheme compared to the hydroclimatological model. The hydroclimatological model requires seasonally-varying parameters to reflect the variations in the impact of hydrological and meteorological conditions on stream temperature. The seasonal-varying parameters increase the model complexity and model calibration effort. The equilibrium temperature model uses the initial stream temperature and wind speed to calculate the initial value of $K_T$ (heat exchange coefficient). Moreover, air temperature and solar radiation are used to calculate the equilibrium temperature, which is the critical variable for water-air heat transfer process. Therefore, the $K_T$ and equilibrium temperature vary temporally and spatially as the meteorological input data, such as air temperature, solar radiation and wind speed, vary. Figure 3 illustrates temporal and spatial variations of monthly average $K_T$ values in the subbasins of upper, middle and downstream of the ARB. It can be seen that the averaged $K_T$ had an obvious seasonal variation for all three subbasis in different parts of the ARB and spatial variations. As a result, the equilibrium temperature model does not need temporal and spatial varying parameters, which reduces the model complexity and calibration efforts. The results showed that the equilibrium temperature model had consistent simulation performances (with NSE values all greater than 0.67) for different stations in different regions of the ARB, which proves the effectiveness of this simple parameterization scheme. In addition, the equilibrium temperature model considers the impact of water depth on the heat transfer process at the water-air interface. Because the change of stream water depth represents the variations of streamflow in the reach, the equilibrium temperature model can simulate the impact of variations in hydrological conditions (representing by streamflow) on the water-air heat transfer processes by incorporating the water depth. As the water depth was incorporated in the stream temperature simulation, the potential uncertainties associated with the algorithms used in SWAT for calculating stream characteristics like stream width could impact the stream temperature simulation and need further investigation. To analyze the sensitivity of water depth impact on stream temperature simulation, the simulation at Athabasca River at Old Fort station was used and the results showed that a 50% increase for water depth would decrease the temperature by 0.28 °C and a 50% decrease for water depth would increase the temperature by 0.33 °C.

For further testing, model calibration and validation for the equilibrium temperature model were also performed on the North Fork Tolt River Watershed. The calibration and validation for streamflow and stream temperature using the original SWAT and hydroclimatological model has been previously accomplished by Ficklin et al. (2012). The SWAT model had a satisfactory performance for streamflow simulation with NSE values as 0.65 and 0.57 during calibration and validation period (see Ficklin et al., 2012 for detailed information). The parameters of equilibrium temperature model were calibrated using observed stream temperature data. η, Lag, λ and $\lambda_{kt}$ were calibrated as -2.5, 6, 0.72 and 0.85, respectively. The comparison of model performance is shown in Table 4. The results indicate that the equilibrium temperature model had minimum RE values compared to other two models. The NSE values were much higher than original SWAT model but lower than the hydroclimatological model. Overall, the equilibrium temperature model had a reasonable performance for stream temperature in the North Fork Tolt River Watershed. The lower NSE values of the equilibrium temperature model proposed compared with hydroclimatological model indicate that more model testing needs to be done for different watersheds to verify the model performance.

**Table 4** Calibration and validation statistics for the three stream temperature models for the North Fork Tolt River Watershed

| Model | Calibration | | | Validation | | |
|---|---|---|---|---|---|---|
| | NSE | $R^2$ | RE (%) | NSE | $R^2$ | RE (%) |
| Original SWAT | -1.6 | 0.87 | 43.4 | -1.54 | 0.85 | 43.4 |
| Hydroclimatological model | 0.70 | 0.79 | -7.6 | 0.77 | 0.83 | -6.1 |
| Equilibrium temperature model | 0.41 | 0.69 | 1.0 | 0.47 | 0.76 | -0.6 |

### 3.4 Impact of the stream temperature simulation on water quality modelling

The impacts of the stream temperature simulation on the reaction rates were investigated at first. The reaction rates at 20 °C are used as the input parameter, and the reaction rates for each day are corrected based on equation 13 using the stream temperature simulated by the SWAT model. An inaccurate or uncalibrated stream temperature simulation may lead to uncertainties of the chemical reaction rates for water quality modelling. Four reaction rates that represent stream water quality in SWAT were chosen to analyze the impact of stream temperature simulation on water quality modelling. These reaction rates are related to carbonaceous biochemical oxygen demand (CBOD), dissolved oxygen (DO), nitrogen (N) and phosphorus (P) simulation in the stream (Table 5). The reaction rates at 20 °C and temperature correction coefficients are defined according to the default values in the SWAT manual (Arnold et al., 2013), and the mean values of chemical reaction rates for different stream temperature simulations are given in Table 5. As an example, the monthly average values of BC3 were shown in Figure 4 to illustrate temporal variations under different water temperature models. The results showed that the chemical reaction rates differed in magnitude and temporal variation under different stream temperature simulations, which impacted the chemical concentration simulation within the stream. These differences in the reaction rates were caused by the different stream temperature simulations using the same reaction rates at 20 °C.

**Table 5** Chemical reaction rates and their mean values under the different stream temperature models

| Name | Description | $K_{20}$ (/day) | $\theta$ | Mean value | | |
|---|---|---|---|---|---|---|
| | | | | Original SWAT | Hydroclimatological model | Equilibrium temperature model |
| RK1 | CBOD deoxygenation rate | 1.71 | 1.047 | 1.072 | 0.999 | 1.069 |
| RK2 | Oxygen reaeration rate | 50 | 1.024 | 31.356 | 29.197 | 31.265 |
| BC3 | Organic N hydrolysis rate | 0.21 | 1.047 | 0.132 | 0.123 | 0.131 |
| BC4 | Organic P mineralization rate | 0.35 | 1.047 | 0.220 | 0.204 | 0.219 |

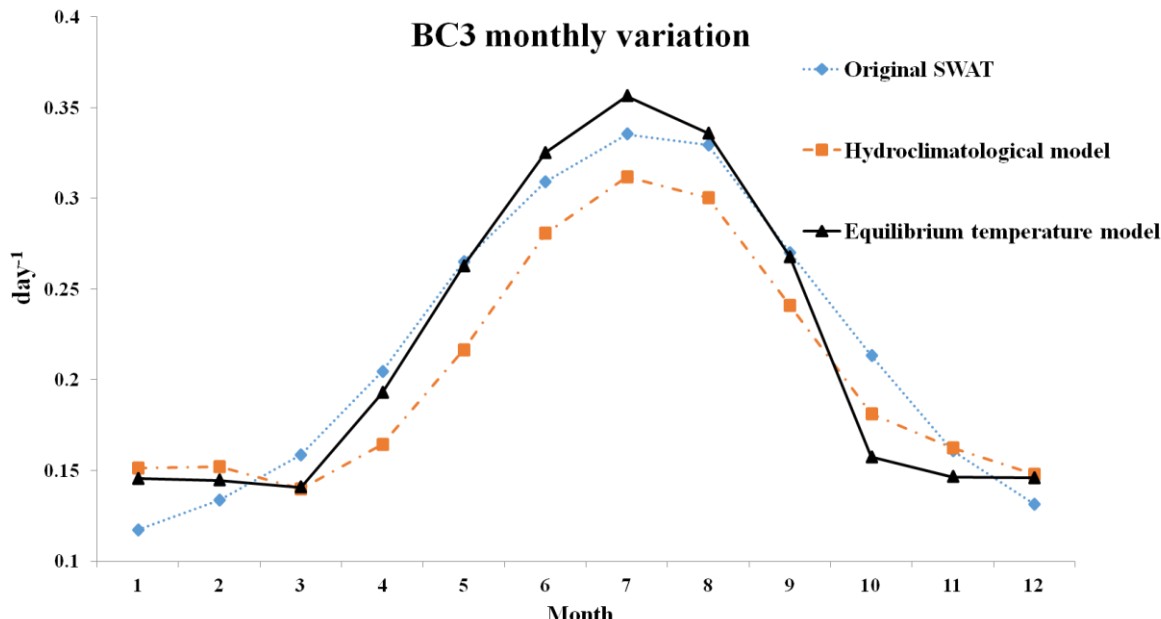

**Figure 4** Monthly variations for the parameter BC3 using the different stream temperature models

To investigate the impact of different stream temperature simulations on water quality concentration simulations, the simulated organic N concentrations for two streams (Table 6) by the three different stream temperature models were analyzed and the average value and standard deviation were selected for assessment. The stations of Muskeg River near Fort Mackay (MRFM) and Athabasca River at Old Fort (AROF) were used for output comparison. The station of MRFM represents an upstream subbasin with no inflow impact, while the station of AROF is in the mainstream of downstream ARB representing the impact of inflow from upstreams. As can be seen from Table 6, annual average organic N concentrations simulated by three different stream temperature models at MRFM were very similar, but the simulated concentrations at AROF showed greater differences. To investigate the impacts on water quality concentration temporal patterns, the monthly average of simulated organic N concentrations from the three different stream temperature models at the MRFM and AROF stations are shown in Figure 5. The

results in Figure 5 indicate that monthly organic N concentrations simulated by the different models showed greater variations at the AROF station, especially from April to June when the concentrations are high. The results implied that the simulations of stream temperature have more impact on the simulated water quality concentrations in the downstream with upstream inflow impact than those located in the

5      upstream with no inflow impact. The simulated daily organic N concentrations of the first year (1988) during simulation period of three different models were outputted and plotted (Figure 6) as an example to show the diurnal variation. The results indicate that the daily concentrations simulated by the hydroclimatological model showed a contrasting pattern compared to other two models as a result of the different stream temperature simulations.

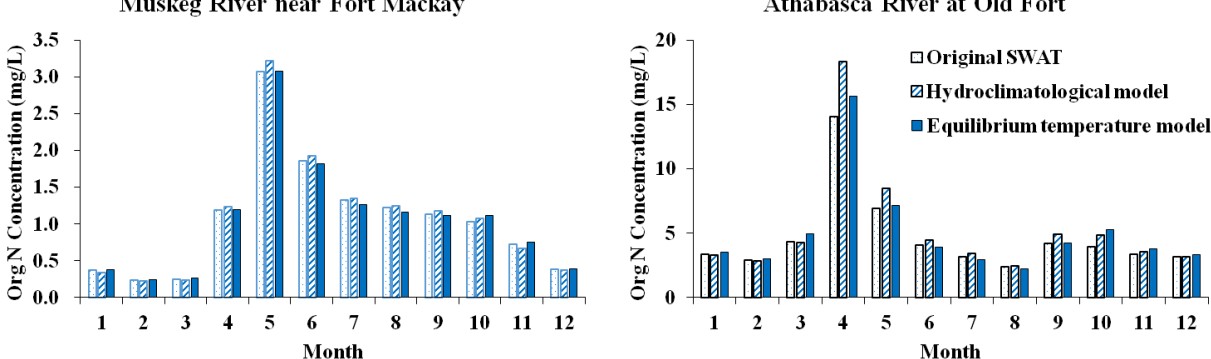

**Figure 5** Monthly variations of organic N concentration simulated using the three different stream temperature models

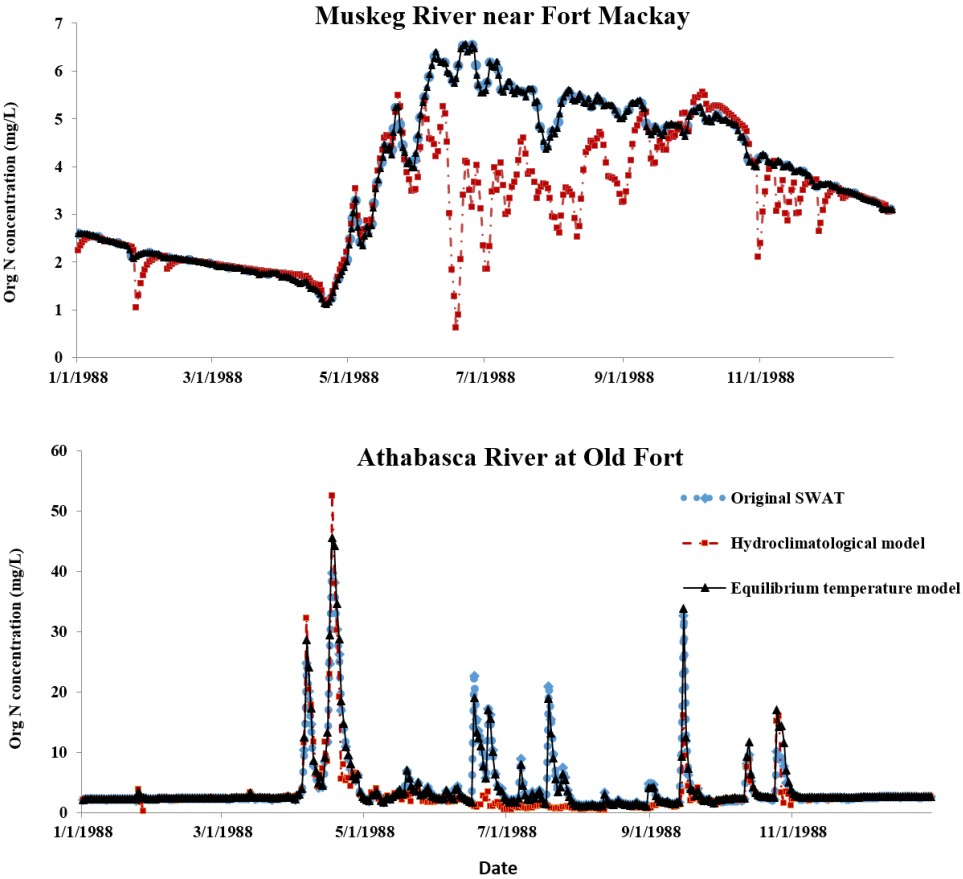

**Figure 6** Daily variations of organic N concentration simulated in 1988 using the three different stream temperature models. *Original SWAT represents the SWAT default linear model, Hydroclimatological model represents the model developed by Ficklin et al. (2010) and equilibrium temperature model represents the model proposed by this study.

**Table 6** Annual average organic N concentrations (mg/L) simulated using the three different stream temperature models

| Station | Statistics | Original SWAT | Hydroclimatological model | Equilibrium temperature model |
|---------|-----------|---------------|---------------------------|-------------------------------|
| Muskeg River near Fort Mackay | Average | 1.064 | 1.087 | 1.063 |
|  | Standard deviation | 1.270 | 1.259 | 1.269 |
| Athabasca River at Old Fort | Average | 4.650 | 5.337 | 4.999 |
|  | Standard deviation | 6.200 | 6.187 | 7.036 |

The SWAT model uses a linear relationship with air temperature to simulate water temperature, and the majority of SWAT applications for water quality modelling do not calibrate and validate stream temperature due to the fixed coefficients for the linear equations. Though water quality concentration can be calibrated by adjusting reaction rates at 20 °C without water temperature calibration, it might not

reflect an accurate representation of chemical reactions and transformations. Therefore, the equilibrium temperature provides a potential tool for more-accurate water quality concentration simulation.

## 4 Conclusions

The temperature of a river system is an important indicator for biodiversity and ecosystem sustainability. The original SWAT model uses a linear equation of air temperature to calculate the stream temperature and does not account for the impact of other meteorological and hydrological conditions. Thus, the linear equation may lead to an unrealistic prediction of the stream temperature when the air temperatures are very high or low. In this paper, we proposed a stream temperature model by incorporating an equilibrium temperature approach to the hydroclimatological model developed by Ficklin et al. (2012). The equilibrium temperature approach accounts for the influence of air temperature, wind speed, solar radiation and water depth to calculate the water-air heat transfer. The hydroclimatological model considers the contribution of different runoff components and calculates the initial stream temperature by mixing runoff from subbasins and inflow from upstream. Then the final stream temperature is calculated by simulating the water-air heat transfer. Compared to the hydroclimatological model, the equilibrium temperature model calculates heat transfer between water and air, including the impact of other meteorological conditions, such as wind speed and solar radiation. Also, the equilibrium temperature model considers the influence of water depth on the heat transfer, which reflects the impact of hydrological variations on water-air heat transfer. An additional advantage of this model is the simple parameterization scheme requiring less calibration effort because it does not need spatial and temporal varying parameters. Also, the equilibrium temperature model uses the existing input data of the SWAT model with no additional inputs.

The equilibrium temperature model was applied to the ARB, and the model calibration and validation were performed using observed stream temperature data from five monitoring stations distributed throughout the ARB. The results show that the equilibrium temperature model had a better performance for the stream temperature simulation than the original SWAT and hydroclimatological models. The equilibrium temperature model showed a consistent performance for different regions in the ARB using fewer parameters and less calibration effort compared to the hydroclimatological model. The equilibrium temperature model was also tested in the North Fork Tolt River Watershed in Washington, United States and the results showed that it had a reasonable performance for stream temperature modeling with minimum RE values compared to other two models but the lower NSE values indicated more model testing needs to be done in future work.

In addition, the impact of the stream temperature on the water quality was analyzed through the variations of chemical reaction rates and concentrations under three different stream temperature models. The results showed the chemical reaction rates and concentrations differed in magnitude and temporal variation under different water temperature simulations, which indicated that stream temperatur simulation is important for water quality modeling. It is worth mentioning that the equilibrium temperature model can also be incorporated to other hydrological models with the required runoff components and meteorological input data. The required meteorological input data includes air temperature, solar radiation and wind speed. The required runoff components consist of surface runoff (overland flow), interflow, groundwater flow and snowmelt. Theoretically, this steam temperature model can be incorporated into any hydrological model used for different regions which has the required metrological and runoff components. Overall, the equilibrium temperature model, which accounts for the combined impact of meteorological and hydrological conditions, can be a useful tool for modelling the stream temperature. The hydroclimatological and equilibrium stream temperature model both use a simple mixing model to calculate the initial stream temperature considering the impact of different runoff

components. This is a simplified simulation for heat process in the subbasin, which can be improved in future studies. In addition, more model testing needs to done in different regions to verify the model applicability. Further work can also be done by incorporating the equilibrium stream temperature model into other hydrologic models for further model testing.

## 5    Acknowledgements

The authors would like to thank the Alberta Economic Development and Trade for the Campus Alberta Innovates Program Research Chair (No. RCP-12-001-BCAIP). We would also like to thank Mr. Jim Sellers for the proofreading. The code of this study is available upon request by email (xdu@athabascau.ca).

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
