# Peer review of "Incorporation of the equilibrium temperature approach in a Soil and Water Assessment Tool hydroclimatological stream temperature model"

_Hydrology and Earth System Sciences, 2017_

## Referee Comment (RC1) · Anonymous Referee #1 · 12 Sep 2017

Review of "Incorporation of the equilibrium temperature approach in a Soil and Water Assessment Tool Hydroclimatological stream temperature model" by Xinzhong Du et al.

In their work, the authors seek to improve the estimation of stream temperature within the SWAT framework by incorporating a model based on the equilibrium temperature approach. By accounting for air temperature, solar radiation, wind speed, and water depth, the authors obtain a more realistic representation of the heat transfer process than the previously used stream temperature models.

The paper is well written and very easy to follow. The authors did a nice job at dis-

cussing the literature and comparing their contribution to previous work. This helped highlight the novelty in their article.

The research is sound and the results brought about by their model, based on the equilibrium temperature approach, are clear. I therefore suggest "minor revision" and only ask the authors to address my few comments below.

Comments:

-Page 3: I agree that the model introduced by the authors represents a trade-off between complex mechanistic models and simple statistical models. However, because of the simplistic representation of the physics, this is an advantage only in long-term analysis, for which complex models would require "intensive data and calibration effort". On the contrary, for short-term analysis a more realistic representation of the physics, as provided by complex mechanistic models, may be more reliable. I think this needs to be briefly discussed in their introduction. By discussing this, the authors would at the same time provide a range of applicability of their model.

-Regarding the organization of the sections, I suggest moving the description of the study area after the description of the model. This would mark more clearly a distinction between the theory (including the novelty of this work) and the application (which mostly has an illustrative purpose).

-Section 2.2, line 15: Saying that the hydrological cycle is simulated based on the water balance is obvious.

-Page 6, equations (4) and (5): can the authors explain why the coefficient of heat transfer should range from 0 to 1?

-Page 7: if the authors do not want to use a model for the dew point temperature (such as Lawrence, 2005), why not calibrating directly Td rather than equating it to Tair+$\eta$ and calibrating $\eta$?

-Figure 2 is too small and needs a higher resolution.

-Why not Arrhenius equation to calculate the change in the reaction rate with respect to temperature? Also, the equilibrium constant too changes with temperature. A more accurate analysis of the effect of temperature on water chemistry would need to account for this (i.e., Van 't Hoff equation).

---

## Referee Comment (RC2) · J. Kiesel (Referee) · 12 Oct 2017

The manuscript "Incorporation of the equilibrium temperature approach in a Soil and Water Assessment Tool hydroclimatological stream temperature model" by Du et al. deals with the modification of the stream temperature simulation in the SWAT model. The work presented in this manuscript builds on previous work by Ficklin et al. 2012. I agree with Reviewer 1 that the paper is generally well-written and structured and easy to follow. However, I see major weaknesses that the Authors should address prior to a possible publication in HESS.

General comments

[Figure]

My major concerns are: - why was the approach only tested in one catchments? Ficklin et al 2012 tested multiple catchments, and is also co-author of this study. Why did you not test your approach in the same SWAT models? - It would have been possible to refrain from using the most sensitive parameter by implementing another equation for depicting dew point temperature. It could be possible that this parameter accounts for other weaknesses in the approach and I therefore ask for additional presentation of intermediate temperature calculations. - I think the comparison to a not calibrate-able equation is unfair (e.g. what if a simple lag factor or multiplicative factor would be included in the original equation?). Most SWAT users will not do or will not be able to calibrate stream temperature. Please provide an assessment using default parameter settings of the equations. - at the current status, the section of the 'sensitivity analysis' regarding water quality is of not much use for the reader. I would like to see a comparison to observations or at least a more detailed presentation of the results (e.g. further statistics based on daily data).

Additional specifications to these concerns and further comments to specific text sections are given below:

Abstract

p.1 l.12-15: At this stage, this is confusing for the reader. Make it clearer that you are modifying the hydroclimatological model

p.1 l.25-26: This is not true: The original model needed no calibration parameter at all.

Introduction

p.1 l.38: ...species have...

p.2 l.4: add industry and power plants

p.2 l.16: I think the equifinality problem does not only apply to statistical methods, but can occur whenever a multi-dimensional parameter space exists, e.g. also for a physically-based model

p.2 l.26-27: This is common sense, I agree, but since you are focusing on those later, is there a reference that lists these parameters as the most influential?

p.2 l.29: I would not generalize it to the point that "statistical models" may not be reliable. I suggest to refer to the previous examples.

p.2 l.36: negligible

p.2 l.42-43: suggest: "...approach. Therefore, these algorithms need to be directly linked or implemented in hydrological models to project the effects..."

p.3 l.5-7: classify the model according to your definition (statistical or mechanical)

p.3 l.7-11: this is repeated later in methods. Suggest to move this detailed information to the methods only.

p.3 l.11-13: This raises the question why the current method is not tested in these seven basins as well. Especially since the author of the paper is also co-Author of this paper

p.3 l.27: ...has rarely been used...

p.3 l.33: The primary objective is not "to develop a stream temperature model". To make it clearer for the reader, I suggest something along these lines: "The primary objective of this paper is to improve the simulation of the heat transfer process in the hydroclimatological stream temperature model on the example of SWAT"

p.3. l.41: ....located in Alberta....

Material and Methods

p.4: The catchment area seems very specific and I am not convinced, that other catchments to test the model are not needed. Please comment.

p.5 l.1: datasets

p.5 l.7-8: I don't understand how you obtain only 1370 HRUs with those numbers of

subbasins, land uses, and soils. What are the (spatial?) pecularities of the setup?

p.5 l.31: I would mention also the case if air temperature suddenly drops or rises

p.6 l.29: It is unclear where the approach is linked to the hydrological stream temperature model. What are the previous equations that are used / replaced or is it added on top of equation 4 and 5? Please mention the link to the equations of the previous chapter.

p.7 l.4: I think you should add "qnet = KT(Te-Tw)"

p.7 l.12-25: I do not understand why you do not use the dew point calculation based on temperature and humidity. Both temp and humidity are SWAT input parameters and the simple equation to calculate dew point does not need calibration. Why did you opt for including an additional calibration parameter? It could be possible that the calibration parameter you include may account for other shortcomings in the model. I suggest to printout an example of the stepwise 'improvement' of temperature depiction vs observed to check the validity of the approach.

p.8 l.19: what was the 'higher sampling frequency'?

p.8 l.24-26: I do not understand...what do you mean with 'one set of parameters were used for the calibration process'?

p.8 l.31-32: Mention which equation and which parameter

Results and Discussion

p.9 l.16: I think even more important than a reasonable NSE for your purpose is the correct depiction of streamflow components (surface runoff, lateral, groundwater flow). Can you comment on the model performance in that regard?

p.9 l.19: considering that this parameter is so sensitive I again need to stress my comment for p.7 l.12-25.

p.9 l.24: I think your comparison is not fair. The current stream temperature model for instance does not need calibration and not every user that depicts water quality has temperature data available to calibrate your model. If this approach will eventually be available in the SWAT model by default, it is extremely important that default parameters are defined that make (some) sense and are applicable for the widest range possible. So, please add a comparison of the three uncalibrated methods.

p.9 l. 21,23 and p.10 l.1: Do you briefly discuss the physical basis / validity for these parameters somewhere? Is it possible to deduce default parameter settings from this?

p.10 l.10-14: Can you discuss why the hydroclimatological model is worse than the original model? It performed so significantly better in Ficklin 2012 in multiple catchments

p.10 l.19: What is so special about that station? Connect the physical catchment properties to the simulations. And why was no validation carried out there?

p.12 l.1: labels on the figure are too small. It is unclear where those subbasins are located in the basin and why they were chosen. Maybe it is better to show box- or violin plots of the 12 months incl uding all subbasins.

p.12 l.16: This is repetition from the methods.

p.12 l.22-24: water depth in swat depends largely on river width...how did you make sure that water depth is reasonable and how sensitive is water depth in the approach?

p.13 l.1: It is uncommom to have an equation in the results. Why didn't you include it to the methods? Unclear to the reader where k20 and teta come from - mention e.g. that the values are SWAT default parameters.

p.13 l.4: Table 4: I think the mean values in the table are misleading. The numbers of the equilibrium model are almost the same as the original SWAT code despite the fact that it performs so much better than the original model. While the hydroclimatological model shows siginificantly different values, though it performs similarly to the original

model. Please consider showing three diagrams similar to Figure 4. For BC3: Probably "Organ N hydolosis rate" is "Organic N..."? Figure 4: Text too small, replace "SWAT" through 'Original SWAT"

p.14 l.19: located

p.14 l.21: Figure 5: You did not compare it to measurements (I would have loved to see it), but are these changes significant and plausible and do they go into the right direction, do they improve the water quality simulation?

p.14 l.22: Table 5: These values do not mean much...e.g. changes in the second digit for average water quality parameter at Muskeg are irrelevant. I suggest to add further statistics: e.g. the standard deviation, 2, 20, 80, 98 percentile based on your daily simulations.

Conclusions

p.15 l.24: looking at figure 2, this seems different. The blue dots are not on a daily time step.

p.15 l.28: Please discuss how applicable the model would be in other regions (humid, temperate, arid) regions. Also mention gaps and weaknesses and room for further work.

will the Code be made available?

---

## Author Comment (AC1) · 10 Nov 2017

General comments: In their work, the authors seek to improve the estimation of stream temperature within the SWAT framework by incorporating a model based on the equilibrium temperature approach. By accounting for air temperature, solar radiation, wind speed, and water depth, the authors obtain a more realistic representation of the heat transfer process than the previously used stream temperature models. The paper is well written and very easy to follow. The authors did a nice job at discussing the literature and comparing their contribution to previous work. This helped highlight the novelty in their article. The research is sound and the results brought about by their

model, based on the equilibrium temperature approach, are clear. I therefore suggest "minor revision" and only ask the authors to address my few comments below.

Response: We would like to thank the reviewer for his/her thoughtful comments. We responded to your specific comments one-by-one below.

Comments 1: Page 3: I agree that the model introduced by the authors represents a trade-off between complex mechanistic models and simple statistical models. However, because of the simplistic representation of the physics, this is an advantage only in long-term analysis, for which complex models would require "intensive data and calibration effort". On the contrary, for short-term analysis a more realistic representation of the physics, as provided by complex mechanistic models, may be more reliable. I think this needs to be briefly discussed in their introduction. By discussing this, the authors would at the same time provide a range of applicability of their model.

Response to Comments 1: Thank very much for your thoughtful comments. We agree with you that complex mechanistic models might be more reliable by more realistically representing the physical processes despite the fact that they require more intensive data and calibration effort. A brief discussion for this point was added in the manuscript in the section of Introduction—"On the other hand, complex mechanistic models might be more reliable by more realistically representing the physical processes compared to statistical models. Therefore, the equilibrium temperature approach, which can be seen as a compromise between an empirical statistical and a complex mechanistic model, can be used as an alternative for simulating the heat transfer processes." The discussion of the applicability of the model was added in the section of Conclusion (Line 16-19, Page 16) as "Theoretically, the equilibrium stream temperature proposed in the manuscript can be incorporated in any other hydrological model with the required runoff components and meteorological input data. The required meteorological input data includes air temperature, solar radiation and wind speed. The required runoff components consist of surface runoff (overland flow), interflow, groundwater flow and snowmelt. "

Comments 2: Regarding the organization of the sections, I suggest moving the description of the study area after the description of the model. This would mark more clearly a distinction between the theory (including the novelty of this work) and the application (which mostly has an illustrative purpose).

Response to Comments 2: Thank you very much for your useful comments. The description of the study area was moved after the description of the model theory to reorganize the section of 'Materials and Methods'. In this way, it marks more clearly a distinction between the theory and the application.

Comments 3: Section 2.2, line 15: Saying that the hydrological cycle is simulated based on the water balance is obvious.

Response to Comments 3: Thank you very much for your useful comments. The expression for "hydrological cycle is simulated based on the water balance" was deleted. Also, the sentence was revised as "The simulated processes of the hydrological cycle in SWAT include canopy interception, surface runoff, infiltration, lateral flow, snowmelt flow, evapotranspiration, deep percolation, groundwater flow and water routing in the stream and other water bodies."

Comments 4: Page 6, equations (4) and (5): can the authors explain why the coefficient of heat transfer should range from 0 to 1

Response to Comments 4: K is a bulk coefficient of heat transfer and ranges from 0 to 1 in equation 4 and 5. The value of K is dependent on the relationship between stream and air temperature within a subbasin. For example, if stream temperature is approximately the same as air temperature, then K is 1. If there is a short travel time or extensive tree shading, then K will be less than 1 but greater than 0. K as 0 means there is no heat transfer between air and water. The above information was added to the manuscript to explain the reason why the coefficient of heat transfer ranges from 0 to 1.

Comments 5: Page 7: if the authors do not want to use a model for the dew point temperature (such as Lawrence, 2005), why not calibrating directly Td rather than equating it to Tair+$\eta$ and calibrating $\eta$?

Response to Comments 5: Thank you very much for thoughtful comments. The major consideration here is that dew point is not an input to the SWAT model and using dew point as input data will increase the input data requirement of the existing SWAT model. Our goal in this manuscript is to incorporate the equilibrium temperature approach into the SWAT hydroclimatological stream temperature model using the existing input weather data such as air temperature, wind speed and solar radiation. Therefore, dew point Td is not directly used as model input. Moreover, dew point can be estimated by air temperature based a linear equation using the additive manner (Lawrence, 2005). In addition, the equilibrium temperature can also be calculated using air temperature instead of the dew point temperature (Dingman, 1972). So, air temperature and an additive parameter (Tair+$\eta$) are used to replace the dew point temperature to estimate the equilibrium temperature. Reference: Lawrence, M. G.: The relationship between relative humidity and the dewpoint temperature in moist air - A simple conversion and applications, B Am Meteorol Soc, 86, 225-+, 2005. Dingman, S. L.: Equilibrium Temperatures of Water Surfaces as Related to Air Temperature and Solar-Radiation, Water Resour Res, 8, 42-&, 1972

Comments 6: Figure 2 is too small and needs a higher resolution.

Response to Comments 6: Figure 2 was revised to have a bigger size and higher resolution according to your comment.

Comments 7: Why not Arrhenius equation to calculate the change in the reaction rate with respect to temperature? Also, the equilibrium constant too changes with temperature. A more accurate analysis of the effect of temperature on water chemistry would need to account for this (i.e., Van't Hoff equation).

Response to Comments 7: The impact of stream temperature simulated by the different

models was analyzed using SWAT's exponential correction equation. The temperature correction equation (equation 12 in the manuscript) used in SWAT model is actually the Arrhenius rate function. As the model used in the manuscript is SWAT model, we used SWAT's Arrhenius rate function to analyze the impact of different stream temperature simulations on water quality simulation. Since the stream temperatures simulated by different models are different, the reaction rates would show differences under other correction functions such as the Van't Hoff equation. However, we focused on using SWAT's Arrhenius rate function to investigate the impacts.

---

## Author Comment (AC2) · 10 Nov 2017

General comments: My major concerns are: - why was the approach only tested in one catchments? Ficklin et al 2012 tested multiple catchments, and is also co-author of this study. Why did you not test your approach in the same SWAT models? - It would have been possible to refrain from using the most sensitive parameter by implementing another equation for depicting dew point temperature. It could be possible that this parameter accounts for other weaknesses in the approach and I therefore ask for additional presentation of intermediate temperature calculations. - I think the comparison to a not calibrateable equation is unfair (e.g. what if a simple lag factor or multiplicative

factor would be included in the original equation?). Most SWAT users will not do or will not be able to calibrate stream temperature. Please provide an assessment using default parameter settings of the equations. - at the current status, the section of the 'sensitivity analysis' regarding water quality is of not much use for the reader. I would like to see a comparison to observations or at least a more detailed presentation of the results (e.g. further statistics based on daily data).

Response to general comments:

General comments 1: why was the approach only tested in one catchments? Ficklin et al 2012 tested multiple catchments, and is also co-author of this study. Why did you not test your approach in the same SWAT models?

Response to general comments 1: Thank you very much for your comments. We agree that a new model should be tested and verified under different conditions. Actually, the study area (Athabasca River Basin) in this manuscript is a not a small specific catchment, but a large river basin with the area as 159000 km2. In general, Athabasca River Basin can be divided into five different regions, namely headwaters, foothill, Prairie, Lesser Slave and boreal (Shrestha et al., 2017), which are associated with different characteristics such as metrological condition and land covers. The five stations used for stream temperature calibration are spatially-varied throughout the Athabasca River Basin (from upstream to downstream) in different sub-regions. Therefore, this study is, in practice, calibrated using five catchments. Although the more calibrations may be better, we believe that the five selected observed stations in different sub-regions are representative of different metrological, hydrological and land cover conditions. The co-author's study (Ficklin et al 2012) used the data of seven watersheds in adjacent regions to verify the hydroclimatological stream temperature model. The areas of those seven watersheds range from 27 to 3354 km2, which are very small compared to Athabasca River Basin. So, testing the new model in five stations throughout large river basins like Athabasca River Basin in different sub-regions is similar to testing in five different watersheds like the co-author's study in 2012. We believe that model

testing in this stage is sufficient as an initial application and verification. Therefore, this paper is self-contained and more testing may distract readers from the development of the model itself, which is a focus of this study. Moreover, future work using this model should be tested on watersheds with different hydrological and environmental conditions. Reference: Shrestha, N. K., Du, X., and Wang, J.: Assessing climate change impacts on fresh water resources of the Athabasca River Basin, Canada, The Science of the total environment, 601-602, 425-440, 10.1016/j.scitotenv.2017.05.013, 2017 Ficklin, D. L., Luo, Y. Z., Stewart, I. T., and Maurer, E. P.: Development and application of a hydroclimatological stream temperature model within the Soil and Water Assessment Tool, Water Resour Res, 48, Artn W0151110.1029/2011wr011256, 2012.

General comments 2: It would have been possible to refrain from using the most sensitive parameter by implementing another equation for depicting dew point temperature. It could be possible that this parameter accounts for other weaknesses in the approach and I therefore ask for additional presentation of intermediate temperature calculations.

Response to general comments 2: Thank you very much for your comments. We used air temperature and an additive parameter ($T_{air}+\eta$) to replace dew point temperature for the heat transfer calculation for three main reasons. Firstly, dew point temperature is much more difficult to obtain as a meteorological input than air temperature (which is also not part of the existing SWAT input data). Using dew point temperature as input parameter would hinder the application of the equilibrium temperature model by requiring additional input data in SWAT model. Secondly, dew point temperature can be calculated by air temperature and relatively humidity using a simple linear equation (Lawrence, 2005) and the equation format is similar to $T_{air}+\eta$. However, this equation has a limitation, which is only fairly accurate for relative humidity values above 50%. For more general conditions, we used air temperature and an additive parameter ($T_{air}+\eta$) to replace dew point temperature. Thus, the users can calibrate this additive parameter to for their study areas instead of using the same equation and coefficient to calculate the dew point temperature. While this might be inconvenient, this provides the users

an approach for their own conditions t, which is important to extend the range of the equation to areas where humidity of below 50%. Thirdly, air temperature and an additive parameter (Tair+$\eta$) were used to calculate equilibrium temperature in equation 11 in the original manuscript, which is an important variable for heat transfer process. So, the equilibrium temperature is linearly related to air temperature in our model, which is consist with other studies (Caissie et al., 2005; Bustillo et al., 2013) using the equilibrium temperature approach for water temperature modeling. Thus, $\eta$ is the additive parameter representing the linear relationship between air temperature and the equilibrium temperature and it can be calibrated by comparing the simulated and observed stream temperature. In conclusion, we think that it's practical and reasonable to use air temperature and an additive parameter (Tair+$\eta$) to replace dew point temperature for heat transfer calculation in the equilibrium temperature model. We add the further information to clarify why air temperature and an additive parameter is better than dew temperature for calculating heat transfer process in the revision (Line 30 Page 6 to Line 5 Page 7). Reference: Lawrence, M. G.: The relationship between relative humidity and the dew point temperature in moist air - A simple conversion and applications, B Am Meteorol Soc, 86, 225-+, 2005. Caissie, D., Satish, M. G., and El-Jabi, N.: Predicting river water temperatures using the equilibrium temperature concept with application on Miramichi River catchments (New Brunswick, Canada), Hydrol Process, 19, 2137-2159, 2005. Bustillo, V., Moatar, F., Ducharne, A., Thiéry, D. and Poirel, A., 2014. A multimodel comparison for assessing water temperatures under changing climate conditions via the equilibrium temperature concept: case study of the Middle Loire River, France. Hydrological Processes, 28(3), pp.1507-1524.

General comments 3: I think the comparison to a not calibrateable equation is unfair (e.g. what if a simple lag factor or multiplicative factor would be included in the original equation?). Most SWAT users will not do or will not be able to calibrate stream temperature. Please provide an assessment using default parameter settings of the equations.
Response to general comments 3: The goal of model comparison in this study is to compare the model performance of the equilibrium stream temperature model with the original SWAT and hydroclimatological stream temperature model. As the original SWAT uses a linear equation with coefficients, we used the default coefficients to run the model to obtain the stream temperature simulation and compared it to other two models. Stream temperature is one of the common monitored water quality variables, which is not difficult to mesure. If water quality concentrations (like nutrients) are measured, it's very likely that stream temperature is also observed. We agree with you in specific comment 27 that it is important to define default parameters to make the model more applicable. We give the default paraneter values as follows: $\eta$ as 0, Lag as 3 days, $\lambda$ and $\lambda$kt as 1.0. In the revision, we will provide defaut coefficients. Those default parameters might be used for the users who don't or can't calibrate for stream temperature. However, it's highly recommended to use observed stream temprature data to calibrate the model paramaters instead of using the default values. We think that giving default parameters works for some cases, but not for most cases. In this study, SWAT default parameters provides resonable results because the linear co-efficients of water and air temperatue regression in our basin is very close to SWAT default values and we have done the linear regression to prove this (please refer to our response to specific comment 29). But for the co-author's study, SWAT default values performs poorly for stream temprature simulations. This is likely because the linear coefficients of water and air temperatue regression are very different from the default coefficient values. This also implies that it's nescessary to calibrate the model using observed data instead of using the default value. In our manuscript, we also analyzed the impacts of stream temperature simulation on water quality modeling. To get a reasonable water quality simulation, it's necessary to calibrate the stream temprature using observed data because it impacts chemical reaction rates. For the hydroclimatological stream temperature model, it's difficult to genelize the default parameter settings. You can see the calibrated parameter values in Table 4 in seven different study watersheds in the co-author's study (Ficklin et al 2012). It can seen from the table that the parameters vary among different seasons and watersheds which makes it hard to give default parameter values. Those paraneters are conceptual parameters reflecting physcical characteristics in different watersheds and therefore to the model must be calibrated using observed data.

General comments 4: at the current status, the section of the 'sensitivity analysis' regarding water quality is of not much use for the reader. I would like to see a comparison to observations or at least a more detailed presentation of the results (e.g. further statistics based on daily data).

Response to general comments 4: We agree with you that a further analysis for the impact of stream temperature simulations on water quality modeling using observed water quality concentration would be beneficial. We add further statistics by outputting and comparing the daily simulated water quality concentrations of three different stream temperature models according to your suggestion. In SWAT chemical reaction rates are affected by stream temperature via an exponential equation. Therefore, simulations of stream temperature impact water quality concentrations by directly impacting the reaction rates. If the stream temperature is not well represented and simulated, it would impact and bring uncertainties for water quality modeling. For a static analysis, the water concentration will be different based on the same base reaction rate (rate at 20℃) and different simulated stream temperature series. For a dynamic analysis in a water quality modeling case, the simulated concentrations can be calibrated to match the observed data by changing the base reaction rate regardless the stream temperature simulations. However, the calibrated base reaction rate might not reflect the real value if the stream temperature is not well represented. Our goal in this manuscript is to initially evaluate the impacts of stream temperature simulation on water quality modeling to illustrate the importance of the water temperature simulation to the readers. We added this as one of the objectives in this manuscript in the section of "Introduction" to clarify this goal. However, we believe a better representation of stream temperature is a precondition of good water quality simulations since the temperature impacts the

reaction rate. More analysis needs to done by comparing the water quality simulation efficiency and analyzing model uncertainty using the observed concentration data. We plan to further research this in another manuscript using observed periodic nutrient (nitrogen and phosphorous) concentrations to investigate the impacts of stream temperature simulations on water quality modeling in terms of model efficiency and parameter uncertainty. However, it's out of scope of the current manuscript.

Specific comments and response:

Comment 1: p.1 l.12-15: At this stage, this is confusing for the reader. Make it clearer that you are modifying the hydroclimatological model

Response to Comments 1: Thanks very much for your useful comment. We made a revision here to clearly give the research goal for our manuscript. We revised the sentence as "In this study, we modified the hydroclimatological model by including the equilibrium temperature approach to model the heat transfer processes at the water-air interface, which reflects the influences of air temperature, solar radiation, wind speed and stream water depth on the heat transfer process."

Comment 2: p.1 l.25-26: This is not true: The original model needed no calibration parameter at all.

Response to Comments 2: As regard to fewer parameters and less effort for the equilibrium temperature model, we are making comparison with the hydroclimatological model not with the original model. We revised the manuscript to clarify this Overall, the equilibrium temperature model uses existing SWAT meteorological data as input, can be calibrated using fewer parameters and less effort compared to the hydroclimatological model."

Comment 3: p.1 l.38: ...species have.

Response to Comments 3: Corrected within the manuscript.

Comment 4: p.2 l.4: add industry and power plants

Response to Comments 4: Thanks very much for your useful comments. The impacts of industry and power plants were added in this sentence. It was revised as "Stream temperature regimes have been and will continue to be affected by anthropogenic activities, especially from thermal inputs from industry and power plants, landuse change, and climate change."

Comment 5: p.2 l.16: I think the equifinality problem does not only apply to statistical methods, but can occur whenever a multi-dimensional parameter space exists, e.g. also for a physically-based model

Response to Comments 5: We agree with you that the equifinality problem does not only apply to statistical methods but it does apply to physically-based models such as SWAT. Multiple different parameter sets can result in similar simulation performances.. A widely used approach investigating the equifinality problem in hydrological modeling is GULE (generalized likelihood uncertainty estimation), which has been used for the SWAT model (Shen et al., 2012). By discussing the statistical models in this paragraph of the Introduction, we aimed to explain that statistical models might not be able to assess the impacts of landuse change or hydrological conditions on stream temperature as those factors may not be incorporated in statistical models. With careful considerations, we decided to delete the equifinality discussion. Reference: Shen, Z. Y., L. Chen, and T. Chen. "Analysis of parameter uncertainty in hydrological and sediment modeling using GLUE method: a case study of SWAT model applied to Three Gorges Reservoir Region, China." Hydrology and Earth System Sciences 16, no. 1 (2012): 121.

Comment 6: p.2 l.26-27: This is common sense, I agree, but since you are focusing on those later, is there a reference that lists these parameters as the most influential?

Response to Comments 6: We now add a reference here to explain the factors affecting stream temperature according to your suggestion.

Comment 7: I would not generalize it to the point that "statistical models" may not be

reliable. I suggest to refer to the previous examples.

Response to Comments 7: Thanks for catching this. We didn't mean to generalize that statistical models may not be reliable for stream temperature simulation. We meant to say that statistical models might not be suitable for evaluating the impact of environmental and anthropological drivers like climate and landuse change because the impact of watershed hydrological conditions are not included in these regression models. We revised the sentence to clarify this "Moreover, the impact of watershed hydrological conditions are not included in these regression models (Ficklin et al., 2012). Therefore, the statistical models of stream temperature may not be reliable when interpreting and predicting the impact of environmental and anthropological drivers, such as climate and landuse change."

Comment 8: p.2 l.36: negligible

Response to Comments 8: Corrected in the manuscript.

Comment 9: suggest: "...approach. Therefore, these algorithms need to be directly linked or implemented in hydrological models to project the effects..."

Response to Comments 9: This sentence was revised according to your comment as "Therefore, these algorithms need to be directly linked or implemented in hydrological models to assess the effects watershed hydrological conditions on stream temperature.".

Comment 10: p.3 l.5-7: classify the model according to your definition (statistical or mechanical)

Response to Comments 10: We added the classification for the hydroclimatological model in the manuscript as "It is a mechanistic model with a simplified representation of temperature mixing from different runoff components and water-air heat transfer processes."

Comment 11: p.3 l.7-11: this is repeated later in methods. Suggest to move this

detailed information to the methods only.

Response to Comments 11: Thanks very much for your useful suggestion. This information was deleted in the section of "Introduction" and has been moved to "Materials and Methods".

Comment 12: p.3 l.11-13: This raises the question why the current method is not tested in these seven basins as well. Especially since the author of the paper is also co-Author of this paper.

Response to Comments 12: Thanks very much for this comment. Please refer to the response to general comment 1.

Comment 13: p.3 l.27: ...has rarely been used...

Response to Comments 13: Corrected within the manuscript.

Comment 14: p.3 l.33: The primary objective is not "to develop a stream temperature model". To make it clearer for the reader, I suggest something along these lines: "The primary objective of this paper is to improve the simulation of the heat transfer process in the hydroclimatological stream temperature model on the example of SWAT"

Response to Comments 14: Thanks very much for your useful suggestion. We modified the primary objective as "improve the simulation of the heat transfer process in SWAT hydroclimatological stream temperature model by incorporating the equilibrium temperature approach"

Comment 15: p.3. l.41: ....located in Alberta...

Response to Comments 15: Corrected within the manuscript.

Comment 16: p.4: The catchment area seems very specific and I am not convinced, that other catchments to test the model are not needed. Please comment.

Response to Comments 16: Please refer to the response to general comment 1.

Comment 17: p.5 l.1: datasets

Response to Comments 17: Corrected within the manuscript.

Comment 18: p.5 l.7-8: I don't understand how you obtain only 1370 HRUs with those numbers of subbasins, land uses, and soils. What are the (spatial?) pecularities of the setup?

Response to Comments 18: The HRUs were defined based on landuse type, soil type and slope classes. To define the HRUs, slope map is derived from the DEM and is divided into 4 classes (breaks at 5%, 10%, 15% and 20%). Moreover, a threshold of 10%, 5% and 10% for land use, soil and slope, respectively, were used for defining HRUs, which resulted in a total of 1370 HRUs in the whole Athabasca River Basin. The above information was added in the manuscript to describe how the HRUs were setup. In addition, more information about SWAT model setup in Athabasca River Basin can refer to our recent publication in "Science of Total Environment" (Shrestha et al., 2017). Reference: Shrestha, N. K., Du, X., and Wang, J.: Assessing climate change impacts on fresh water resources of the Athabasca River Basin, Canada, The Science of the total environment, 601-602, 425-440, 10.1016/j.scitotenv.2017.05.013, 2017

Comment 19: p.5 l.31: I would mention also the case if air temperature suddenly drops or rises

Response to Comments 19: Thanks very much for your useful suggestion. This information was added in the manuscript as "Moreover, it might provide unrealistic simulations when air temperature suddenly drops or rises."

Comment 20: p.6 l.29: It is unclear where the approach is linked to the hydrological stream temperature model. What are the previous equations that are used / replaced or is it added on top of equation 4 and 5? Please mention the link to the equations of the previous chapter.

Response to Comments 20: There are three steps for simulating stream temperature

in the hydroclimatological model. The first two steps calculate the initial stream temperature and the third step calculates the heat transfer in the water-air interface. In this study, the same equations of the first two steps in the hydroclimatological model were used and the equilibrium temperature approach was incorporated to simulate the heat transfer process. So, equations 1 to 3 from hydroclimatological model were used in the modified model. The above information was added to the manuscript to clarify the linkage between the hydroclimatological model and modified model.

Comment 21: p.7 l.4: I think you should add "qnet = KT(Te-Tw)"

Response to Comments 21: An additional equation was added in the manuscript.

Comment 22: p.7 l.12-25: I do not understand why you do not use the dew point calculation based on temperature and humidity. Both temp and humidity are SWAT input parameters and the simple equation to calculate dew point does not need calibration. Why did you opt for including an additional calibration parameter? It could be possible that the calibration parameter you include may account for other shortcomings in the model. I suggest to printout an example of the stepwise 'improvement' of temperature depiction vs observed to check the validity of the approach.

Response to Comments 22: Thanks very much for this comment. Please refer to the response to general comments 2.

Comment 23: p.8 l.19: what was the 'higher sampling frequency'?

Response to Comments 23: There are different observation stations measuring stream temperature and the sampling frequencies of stream temperature varied from monthly to seasonal. The higher sampling frequency here meant that those stations with sampling frequency close to monthly were chosen for model calibration and validation. We revised the manuscript to make this more clear. Moreover, the specific numbers of stream temperature samples for the selected 5 stations can be found in Table 2.

Comment 24: p.8 l.24-26: I do not understand...what do you mean with 'one set of

parameters were used for the calibration process'?

Response to Comments 24: By saying this, we meant that one set of parameters were used for different seasons and subbasins. In other words, we didn't use seasonal and spatial varying parameters for the calibration process. We added this additional information for clarification in the manuscript.

Comment 25: p.8 l.31-32: Mention which equation and which parameter

Response to Comments 25: The detailed information for parameter calibration of hydroclimatological stream temperature model can be found in Table 2. In this study, three different seasons were defined based on the Julian days and different parameters were used for different periods.

Comment 26: p.9 l.16: I think even more important than a reasonable NSE for your purpose is the correct depiction of streamflow components (surface runoff, lateral, groundwater flow). Can you comment on the model performance in that regard?

Response to Comments 26: We agree that correct depiction of runoff components including surface runoff, lateral and groundwater flow is very important for hydrology model verification. Streamflow is generated from the landscape via routing processes by different runoff components which all contribute the streamflow. Therefore, a better matching for streamflow may not ensure a reasonable water balance simulation. However, it's difficult to do directly calibrate using these runoff components as they are usually not measured in a watershed scale. Since streamflow is calibrated in another paper published in "Science of Total Environment" (Shrestha et al., 2017), annual water balance during the simulation period and future climate change scenarios were analyzed. Here, we cite a table from that paper to demonstrate the water balance simulated by SWAT in Athabasca River Basin. You can see the water balance components of the simulated period (1983-2013) as "base period". From the table, the results show that the SWAT model performs a reasonable water balance in terms of the ratios of streamflow to precipitation and surface runoff and sub-surface runoff according

to SWAT-Check summary. Table 1 Yearly water balance components for base period (1983-2013) and future periods (2040's and 2080's)

Reference: Shrestha, N. K., Du, X., and Wang, J.: Assessing climate change impacts on fresh water resources of the Athabasca River Basin, Canada, The Science of the total environment, 601-602, 425-440, 10.1016/j.scitotenv.2017.05.013, 2017

Comment 27: p.9 l.24: I think your comparison is not fair. The current stream temperature model for instance does not need calibration and not every user that depicts water quality has temperature data available to calibrate your model. If this approach will eventually be available in the SWAT model by default, it is extremely important that default parameters are defined that make (some) sense and are applicable for the widest range possible. So, please add a comparison of the three uncalibrated methods.

Response to Comments 27: Please refer to the response to general comments 3.

Comment 28: p.9 l. 21,23 and p.10 l.1: Do you briefly discuss the physical basis / validity for these parameters somewhere? Is it possible to deduce default parameter settings from this?

Response to Comments 28: As to the physical basis and meanings of the parameters in the hydroclimatological model, these the co-author's study (Ficklin et al 2012). For the parameters in table 2, K is bulk when coefficient of heat transfer and ranges from 0 to 1, which is dependent on the relationship between stream and air temperature within a subbasin. $\lambda$ is a calibration coefficient relating the relationship between $T_{air,lag}$ and surface runoff and lateral flow. is the additive parameter allows the modeled water temperature to rise above 0℃when the air temperature is below 0. The lag (days) is a calibration parameter incorporated to allow the effects of delayed surface runoff and soil water flow into the stream. As to the parameters in the equilibrium temperature model, $\eta$ is the parameter representing the linear relationship between air temperature and the equilibrium temperature. $\lambda_{kt}$ is the parameter representing the linear ratio of the KT value calculated by the empirical equation (equation 8 in the original manuscript) to

the value in the applied watershed. In addition, lag is also used in our mode which has the same meanings as in the hydroclimatological model. We added the above discussion about parameter meanings for the equilibrium stream temperature model in the manuscript. Ideally, the model parameters can be deduced and no parameter calibration is required. However, this is not the case for most of the watershed models which usually have some empirical or conceptual parameters. For example, the SWAT model has a lot of empirical or conceptual parameters that need to be calibrated using the observed data in the study area. There are several reasons for this; firstly, the empirical or conceptual parameters are an abstract or simplification of the physical processes that cannot be deduced. Secondly, because of the scale issue, a parameter obtained in the lab or field might not be transferable to the watershed scale. Therefore, a parameter calibration process is required for model application. There is one parameter, however, that can be deduced in our model which is the lag (days). It can be obtained by calculating the correlation coefficient between the observed stream temperature and the moving average of air temperature before the day water temperature is measured. For example, if the observed daily stream temperature has the maximum correlation with 3 days average air temperature before the stream temperature is measured, then the initial value of lag can be set as 3 days.

Comment 29: p.10 l.10-14: Can you discuss why the hydroclimatological model is worse than the original model? It performed so significantly better in Ficklin 2012 in multiple catchments.

Response to Comments 29: Thanks very much for your useful comments. Form the statistics in Table 3, it looks like that the original SWAT model has a better performance. However, the simulation results of original SWAT have some abnormal results which the hydroclimatological model doesn't have. For example, the simple linear equation in original SWAT model may lead to unrealistic estimates of stream water temperature when the air temperature is low during winter. Moreover, it might provide unrealistic simulations when air temperature suddenly drops or rises. We have performed a linear

regression between observed stream temperature and air temperature using the data in Athabasca River Basin. It turned out that the linear coefficient (0.76) and interception (5.7) of the linear regression is very close to the SWAT default values (0.75 and 5.0). Thus, SWAT original linear equation could perform reasonable results in Athabasca River Basin. However, for the co-author's study, it might be because the linear coefficients and interceptions of stream temperature and air temperature regression are very different from SWAT default values, which led to a poor performance in those watersheds. The reason for the hydroclimatological model performing not as good in Athabasca River Basin may be because it needs spatially varied parameters to be calibrated for different stations but this study used one set of parameters for the whole Athabasca River Basin to verify the model performance with less calibration effort.

Comment 30: p.12 l.1: labels on the figure are too small. It is unclear where those subbasins are located in the basin and why they were chosen. Maybe it is better to show box- or violin plots of the 12 months including all subbasins

Response to Comments 30: We have revised the figure by enlarging the labels in the figure. Sorry for the confusion because we used the subbasin number of the SWAT model in the Figure. The three subbasins are where the three observed stream temperature stations are located (Athabasca River near Windfall (subbasin 104), Athabasca River at Athabasca (subbasin 97) and Athabasca River at Old Fort (subbasin 5) – located upstream, mid-stream, and downstream, respectively. We revised the figure by using the station name as the labels to avoid the confusion about the location of the three subbasins. These three subbasins were chosen to represent different subregions in Athabasca River Basin because this figure is to show the temporal and spatial variation of KT (heat exchange coefficient). We think that the selected three typical subbasins in Athabasca River Basin in this figure are sufficient to show the spatial variation of KT.

Comment 31: p.12 l.16: This is repetition from the methods.

Response to Comments 31: This sentence was deleted in the manuscript.

Comment 32: p.12 l.22-24: water depth in swat depends largely on river width...how did you make sure that water depth is reasonable and how sensitive is water depth in the approach?

Response to Comments 32: Yes, water depth in SWAT depends on geometry characteristics (such as stream width, slope and cross-sectional area) and flow conditions. SWAT model obtains those geometry parameters from DEM analysis using ArcSWAT during model setup. Usually, those parameters are not subject to model calibration and the default values are used. Moreover, SWAT assumes the main channels or reaches have a trapezoidal shape to calculate geometry characteristics such as water depth, velocity and cross-sectional area. Since SWAT is a hydrological model with simplified stream geometry representation, streamflow data rather than water depth is usually used for model calibration and validation. However, once the streamflow is calibrated using the observed data, the flow condition in the stream is verified. Then, water depth is justified based on the flow condition and geometry characteristics. It's worth mentioning that including water depth in the heat transfer process calculation is a theoretical improvement which implicitly considers the impact of hydrology condition on heat transfer processes. For example, lower water depth caused by lower discharges mean lower thermal capacity of streamflow which means stream temperature changes more quickly than a higher water depth.

Comment 33: p.13 l.1: It is uncommom to have an equation in the results. Why didn't you include it to the methods? Unclear to the reader where k20 and teta come from - mention e.g. that the values are SWAT default parameters.

Response to Comments 33: Thanks very much for your useful comments. We moved this equation to the section of "Materials and Methods". In addition, the parameter values of k20 and $\theta$ are defined according to the default values in the SWAT manual which is mentioned in our original manuscript (Line 13-14, Page 13) as "The reaction

rates at 20 °C and temperature correction coefficients are defined according to the default values in the SWAT manual (Arnold et al., 2013)"

Comment 34: p.13 l.4: Table 4: I think the mean values in the table are misleading. The numbers of the equilibrium model are almost the same as the original SWAT code despite the fact that it performs so much better than the original model. While the hydroclimatological model shows siginificantly different values, though it performs similarly to the original model. Please consider showing three diagrams similar to Figure 4. For BC3: Probably "Organ N hydolosis rate" is "Organic N..."? Figure 4: Text too small, replace "SWAT" through 'Original SWAT"

Response to Comments 34: The mean values presented here is the annual average reaction rate under the different stream temperature simulations. We think that the reaction rates under different temperature simulations are not related to the model performance by three different models. We want to show the impacts of stream temperature simulations on the reaction rate magnitudes using the annual average value in this Table. Even though the annual average values of reaction rates under the original SWAT and equilibrium model is pretty close, the temporal variation of the reaction rate is very different. You can see from Figure 4 that the average values of original SWAT and equilibrium model in each month are different. Figure 4 shows the temporal variation of chemical reaction rates by showing average values of BC3 (Organic N hydrolysis rate) in each month. You can see from the exponential correction function for reaction rate that the variation of reaction rates is only caused by different stream temperature simulations as k20 and $\theta$ are two constant coefficients as input parameters. Therefore, the temporal variations of different reaction rates are exactly the same even though the magnitudes are different. Here, we chose BC3 to show the temporal variations of reaction ration caused by different stream temperature simulations. We corrected the spelling mistake as "Organic N hydrolysis rate" in Table 4. We revise Figure 4 to enlarge the text in the figure and using "Original SWAT" to replace "SWAT" in the figure label.

[Figure]

Comment 35: p.14 l.19: located

Response to Comments 35: Corrected within the manuscript.

Comment 36: p.14 l.21: Figure 5: You did not compare it to measurements (I would have loved to see it), but are these changes significant and plausible and do they go into the right direction, do they improve the water quality simulation?

Response to Comments 36: Thanks very much for this useful comment. Please refer to the response to general comment 4.

Comment 37: p.14 l.22: Table 5: These values do not mean much...e.g. changes in the second digit for average water quality parameter at Muskeg are irrelevant. I suggest to add further statistics: e.g. the standard deviation, 2, 20, 80, 98 percentile based on your daily simulations.

Response to Comments 37: In Table 5, we compared the simulated annual average concentrations under different stream temperature models using the same parameters (k20 and $\theta$). According to your suggestion, the standard deviations were added to the table in addition to annual average values. It's worth mentioning that the temporal distributions of simulated concentrations under different stream temperature models showed contrasting patterns, which can be seen in Figure 5.

Comment 38: p.15 l.24: looking at figure 2, this seems different. The blue dots are not on a daily time step.

Response to Comments 38: The black lines in Figure 2 are the simulated continuous daily stream temperatures, while the blue dots are the observed period daily average temperature (not continuous time series data) collected from Environment Canada as mentioned in the manuscript. Usually, unlike the observed streamflow data, stream temperature and other water quality concentrations are not measured continuously, but they are measured periodically (monthly or weekly). The frequencies of observed stream temperatures used in this study were listed in Table 1 in the manuscript. To

clarify this, we use the term "periodic daily stream temperature data" replacing "stream temperature data" in the manuscript.

Comment 39: p.15 l.28: Please discuss how applicable the model would be in other regions (humid, temperate, arid) regions. Also mention gaps and weaknesses and room for further work.

Response to Comments 39: Theoretically, this steam temperature model can be incorporated into any hydrological model which has the required metrological and runoff components in any region. In other words, this stream temperature model is a plug-in module that could be incorporated into any hydrological model used in any region. Different regions (humid, temperate, arid) have different runoff generation mechanisms and various hydrologic models may be more suitable for these environments than others. The stream temperature model in this paper can be used in any region and linked with a suitable hydrology model if a meteorological inputs and hydrologic outputs are available. However, this stream temperature model still needs to be applied in different regions. We add the discussion for model applicability in the section of "Conclusion". The hydroclimatological and equilibrium stream temperature model both use a simple mixing model to calculate the initial stream temperature considering the impact of different runoff components. It's a simplified simulation for the heat processes occurring within a subbasin, which can be improved in future work. Further work can also be done by incorporating the equilibrium stream temperature model into other hydrology modesl for further model testing. We added the above discussion in the section of "Conclusion".

Comment 40: will the Code be made available?

Response to Comments 40: The code can be available upon request by Email, and we add this information in the manuscript.

---

## Author Response (AR1)

Dear Editor,

Thank you very much for your useful comments and giving the opportunity to revise our manuscript. According to your useful suggestion, we used one more watershed (North Fork Tolt River Watershed in Washington, United States) to further test the equilibrium model proposed in this manuscript. We added this further model testing in the manuscript and revised our manuscript according to all the reviewers' comments. The revisions made in the manuscript have been marked as italic and red. Moreover, we had responded to the two reviewers' comments point-by-point in the interactive discussion process of this manuscript (https://www.hydrol-earth-syst-sci-discuss.net/hess-2017-443/).

[revised manuscript text omitted]

---

## Referee Report (RR1)

The authors have provided a thoughtful response to reviewers' comments and have properly revised the manuscript. In my previous review, I was already satisfied by the results of the research and I only had minor concerns. After reading the author's reply and the new version of the manuscript, I think the paper has reached a satisfactory quality and can be considered ready for publication.

---

## Author Response (AR3)

Thanks very much for the comments from the editor and reviewer. As the editor commented based on the each comment from the reviewer, we responded all the comments one-by one and the comments from the editor and the reviewer are marked with blue color. Also, we revised the manuscript according to each comment and marked the changes using *red color and italic* in the revised manuscript.

**Comment 1 from reviewer:** Response to comments 29: you mention that the hydroclimatological model requires spatially varied parameters, but you only used one set "to verify the model with less calibration effort" in Athabasca. I wonder what the results would be if you calibrated the hydroloclimatological model as it should have been done. Would the results be similar to the North Fork Tolt River?

**Comment 1 from editor:** Please confirm if you used the same calibration procedures to calibrate the models in the two watersheds. If they are not the same, the authors must provide strong justification why they use different procedures.

**Response to comment 1:** Thank you very much for the editor's and reviewer's comments. Sorry for the confusion for the response from the first round revision. Actually, the same procedures were used to calibrate the stream temperature models in the two different watersheds. For the hydroclimatological model, we used the seasonally varied parameters for both Athabasca River Basin and North Fork Tolt River watershed. Three different periods were defined for Athabasca River Basin (please see Table 2 in the manuscript) and North Fork Tolt River watershed (please see Table 4 from Ficklin et al. (2012)). For the equilibrium temperature model proposed in this study, we did not use seasonal varied parameters in the two different studied watersheds and therefore one set of parameters were used for the whole simulation period. As we discussed in the section of 3.3 (Line 7-16 Page 14), the key parameter $K_T$ (heat exchange coefficient) had an obvious seasonal variation in different parts of Athabasca River Basin (please see Figure 3 in the manuscript). As a result, the equilibrium temperature model does not need temporal varying parameters, which reduces the model complexity and calibration efforts. More information about model calibration processes can be found in the second paragraph of section 2.4.

**Reference:**

Ficklin, Darren L., et al. "Development and application of a hydroclimatological stream temperature model within the Soil and Water Assessment Tool." Water Resources Research 48.1 (2012).

**Comment 2 from reviewer:** Response to comments 32: I do not agree with your explanation. You write that "lower water depth...means stream temperature changes quicker". In SWAT, river width can be orders of magnitude off the real value because the simplified GIS algorithm does not account for local stream characteristics. You can easily check this by overlaying your stream network on Satellite Images. In most cases this is not a significant problem (though it can be for routing and transmission losses). But in your case, if your width is overestimated by a factor of let's say 5, your depth would be underestimated by a similar value. Keeping in mind that you are proposing a new method of stream temperature calculations in SWAT, I think you should know

(a) if your river width is somewhere near reality and if not, correct it in the .rte (which is not complicated: you can cluster your subbasins according to river order and get representative widths for each order and change the .rte accordingly) and

(b) how sensitivity water depth is in your equation.

**Comment 2 from editor:** Please check if the bankful width used the model match reality values. If credible data is not available, the authors should discuss potential uncertainties associated with the algorithms used in SWAT. I understand that correcting the errors associated with the built-in bankful width estimation algorithm within SWAT is beyond the scope of the study, but certain sensitivity test would be helpful to understand the value river width information for accurate water temperature simulation.

**Response to comment 2:** We agree with the reviewer that there are uncertainties about using GIS and DEM to calculate stream characteristics like stream width which will affect water depth and temperature simulations. By including water depth in the stream temperature simulation (equation 11 in the manuscript), we aimed to claim that (Line 20-23 Page 14) '*Because the change of stream water depth represents the variations of streamflow in the reach, the equilibrium temperature model can simulate the impact of variations in hydrological conditions (representing by streamflow) on the water-air heat transfer processes by incorporating the water depth.*' So, we did not try to directly discuss the impact of water depth on stream temperature simulation. Vliet et al. (2011) did global stream temperature simulation by considering the impact of air temperature and streamflow and indicated model performance is improved by introduction of streamflow in the regression model. Also, the sensitivity analysis showed that decrease of streamflow discharge would increases the stream temperature and they also suggested that the large the streamflow, the longer it takes to warm up the water. Therefore, we believe that incorporation of water depth representing the streamflow condition in the model proposed in this study could improve the simulation of stream temperature.

We agree with the editor that correcting the errors associated with the built-in bankfull width estimation algorithm within SWAT is beyond the scope of this study. The streamflow in Athabasca River Basin is well calibrated, which might indicate that the bank width and other stream characteristics were reasonably represented. We added the discussion about potential uncertainties associated with the algorithms for calculating stream width used in SWAT and its impact on stream temperature simulation according to the comment from the editor (Line 23-26 Page 14). Also, a sensitivity analysis for the impact of water depth on stream temperature simulation was done according to the editor's and reviewer's comments (Line 26-29 Page 14). The stream temperature simulation at Athabasca River at Old Fort station was used to do the sensitivity analysis and the results showed that a 50% increase for water depth will decrease the temperature by 0.28 °C and a 50% decrease for water depth will increase the temperature by 0.33 °C.

**Reference:**

Van Vliet, M. T. H., et al. "Global river temperatures and sensitivity to atmospheric warming and changes in river flow." Water Resources Research 47.2 (2011).

**Comment 3 from reviewer:** Response to Comment 38: Sorry, your answer and p.9 l.16-19 is still confusing for me. "blue dots are the observed period daily average (not continuous) ... measured periodically (monthly or weekly)" So how long is an "observed period daily average" continuously measured? I don't think someone is waiting at each gauge for 24hours and takes a sample every hour, but I suspect what you got is 'only' an instantaneous observation at the particular time of the day when the observation took place at max once a month, or? In that case, I think your explanation is misleading. Please rewrite so that the reader knows the true quality of your data. Considering this, I do not agree with your abstract p.1 l.21-22: What you used is not "high frequency observed stream temperature data"

**Comment 3 from editor:** Please better clarify the description of the data used, and change the abstract to reflect the actual quality of the data you used.

**Response to comment 3:** Thanks very much for the editor's and reviewer's comment. Sorry about the confusion of the description of the observed stream temperature data we used in the Athabasca River Basin. We re-checked the observed stream temperature data obtained from Environmental Canada and Climate Change. They were instantaneous observations at the particular time of the day and the frequency of the samplings varied among different stations. We used 'daily' before because SWAT is ran at the daily scale and the simulated stream temperature can be seen as daily stream temperature. We compared the SWAT simulated stream temperature with the observed data available on the same date. We rephrased the description of observed stream temperature data to clarify the confusion. The modification for the description can be found from Line 19-22 Page 9 in the revised manuscript as "*Stream temperature data from Environmental Canada and Climate Change was used for the stream temperature calibration. The sampling frequencies of stream temperature varied from monthly to seasonal, and five stations with close to monthly sampling frequency were chosen for the model calibration.*"Moreover, the abstract (Line 23 Page 1) was changed as "*where close to monthly samplings of stream temperature are available*" to reflect actual quality of the data according to the editor's and reviewer's comments".

**Comment 4 from reviewer:** p.14 l.24 - p.15 l.9: Your model performs worse in the second catchment. This raises the question "Why is this the case"? You don't discuss possible reasons anywhere.

And equally important: why did you choose this particular catchment out of the seven other catchments? Do you expect your model to perform better in the other catchments?

Besides that, in the abstract and conclusion you only mention the RE and hide the NSE, but in the Athabasca where the NSE is higher, you mention the NSE. I think this is not good practice.

**Comment 4 from editor:** Please clarify the performance of different methods. I also found it is difficult to understand the comparative analysis of the performances of different methods. One suggestion is to use "default" to represent the SWAT built-in method, use "Ficklin et al.'s method" to indicate the work by Ficklin et al., and use "this study" to represent the new method proposed in the manuscript.

If NSE or RE is worse for the new method, the authors need to identify strong reasons to justify why the new method is better than the existing ones. If the new method performed less in the second catchment, and the

authors would like justify the new method's advantage to the existing ones, more watershed should be used and the new method should perform better for most cases.

**Response to comment 4:** Thanks very much for the comments from the editor and the reviewer. First, North Fork Tolt River was selected for model further verification in other region out of Athabasca River Basin in the cold region of Canada according to the first round comment from the reviewer. For North Fork Tolt River, the RE values of the model in this study are better than hydroclimatological model from Ficklin et al.'s method but the NSE values are worse indicating further model testing is needed. We agree with the editor and reviewer that more model verification needs to be done for different watersheds in other regions in future work and we now discuss the discrepancy in the revised manuscript (Line 12-14 Page 15). The discrepancies may be because the hydroclimatological model from Ficklin et al.'s method considered the impact of hydrologic processes and air temperature on stream temperature and the model proposed in this study considers the impact of solar radiation, wind speed and streamflow condition on the stream temperature simulation which improved the Ficklin et al.'s method theoretically by adding more factors. However, it still needs more model testing to verify the model performance in different watersheds of different regions.

Additionally, according to the editor's comments, we used different names for different models to clarify the comparative analysis of the performances of different methods in the abstract. Specifically, we used 'equilibrium temperature model proposed in this study' to represent the modified model in this study, 'hydroclimatological stream temperature model' to represent the Ficklin et al.'s method and SWAT default model to represent the SWAT built-in method. According to the reviewer's comment, we added the description of NSE values for North Fork Tolt River and a short discussion in the abstract (Line 29-30 Page 1).

**Comment 5 from reviewer: p.17 Figure 6:** It looks like the original SWAT is exactly below the newly developed model, right? Is this a labelling error? Considering that you built the model on top of the Hydroclimatological model (which is significantly different from the two), this is very confusing. Based on this result, I do not understand your interpretation on p.18, l.8-9 "the equilibrium temp provides a potential tool for more-accurate water quality concentration simulation" Based on these results, I think it is impossible to know which model version is "more-accurate".

Also, the selection of lines and markers in unfortunate. I suggest to remove the markers and use different thicknesses for the two lines that closely ovelap.

**Comment 5 from editor:** I think the purpose of figure 6 is to show that organic N simulation is sensitive to temperature module. Anyway, please add more information in the caption to highlight the key message from this figure. Are they used to show the new method is better, or the authors intend to show the sensitivity of organic N simulation? Also, please improve the legibility, and clearly label the names of the different method, and which one is the method proposed in the manuscript.

**Response to comment 5:** Thank you very much for the comments from the editor and reviewer. The purpose of Figure 6 is to show the organic N simulation is sensitive to different temperature models since the stream temperature affects the reaction rates and to prove temperature simulation is important for water quality

simulation. Our goal is not to compare which model is better in terms of water quality modeling and we clarify the discussion in the section of conclusion according to the reviewer's comment as (Line 34-36 Page 19) "*The results showed the chemical reaction rates and concentrations differed in magnitude and temporal variation under different water temperature simulations, which indicated that stream temperature simulation is important for water quality modeling*." According to the comments from the reviewer, we revised Figure 6 by using different thicknesses for the two lines closely overlapping. Also, we added an annotation in the title of Figure 6 to clarify the three different models according to the editor's comment.

**Minor comments from the reviewer:**

===============

Comment 1: p.8 l.8: The the

Response: It was corrected in the revised manuscript.

Comment 2: p.8 l.10: drainage area of 105 km²

Response: It was corrected in the revised manuscript.

Comment 3: Response to Comment 40: In the "...version4.pdf" I received, I couldn't find the information.

[revised manuscript text omitted]